# Learning Graph Structure from Convolutional Mixtures

## Abstract

Machine learning frameworks such as graph neural networks typically rely on a given, fixed graph to exploit relational inductive biases and thus effectively learn from network data. However, when said graphs are (partially) unobserved, noisy, or dynamic, the problem of inferring graph structure from data becomes relevant. In this paper, we postulate a graph convolutional relationship between the observed and latent graphs, and formulate the graph learning task as a network inverse (deconvolution) problem. In lieu of eigendecomposition-based spectral methods or iterative optimization solutions, we unroll and truncate proximal gradient iterations to arrive at a parameterized neural network architecture that we call a Graph Deconvolution Network (GDN). GDNs can learn a distribution of graphs in a supervised fashion, and perform link prediction or edge-weight regression tasks by adapting the loss function. Since layers directly operate on, combine, and refine graph objects (instead of node features), GDNs are inherently inductive and can generalize to larger-sized graphs after training. We corroborate GDN's superior graph recovery performance using synthetic data in supervised settings, as well as its ability to generalize to graphs orders of magnitude larger that those seen in training. Using the Human Connectome Project-Young Adult neuroimaging dataset, we demonstrate the robustness and representation power of our model by inferring structural brain networks from functional connectivity.

## 1 Introduction

Inferring graphs from data to uncover latent complex information structure is a timely challenge for geometric deep learning (Bronstein et al., 2017) and graph signal processing (Ortega et al., 2018). But it is also an opportunity, since network topology inference advances (Dong et al., 2019; Mateos et al., 2019) could facilitate adoption of graph neural networks (GNNs) even when no input graph is available (Hamilton, 2020). The problem is also relevant when a given graph is too noisy or perturbed beyond what stable (possibly pre-trained) GNN architectures can effectively handle (Gama et al., 2020a). Early empirical evidence suggests that even when a graph is available, the structure could be further optimized for a downstream task (Kazi et al., 2020; Feizi et al., 2013), or else sparsified to boost computational efficiency and model interpretability (Spielman & Srivastava, 2011).

In this paper, we posit a convolutional model relating observed and latent undirected graphs and formulate the graph learning task as a supervised network inverse problem; see Section 2 for a formal problem statement. This fairly general model is motivated by various practical domains outlined in Section 3, such as identifying the structure of network diffusion processes (Segarra et al., 2017; Pasdeloup et al., 2018), as well as network deconvolution and denoising (Feizi et al., 2013). We propose a parameterized neural network model, termed graph deconvolution network (GDN), which we train in a supervised fashion to learn the distribution of latent graphs. The architecture is derived from the principle of algorithm unrolling used to learn fast approximate solutions to inverse problems (Gregor & LeCun, 2010; Sprechmann et al., 2015; Monga et al., 2021), an idea that is yet to be explored in the context of graph structure identification. Since layers directly operate on, combine, and refine graph objects (instead of nodal features), GDNs are inherently inductive and can generalize to graphs of different size. This allows the transfer of learning on small graphs to unseen larger graphs, which has significant implications in domains like social networks and molecular biology (Yehudai et al., 2021). Our experiments demonstrate that GDNs are versatile to accommodate link prediction or edge-weight regression aspects of learning the graph structure, and achieve superior performance

over various competing alternatives. Building on recent models of functional activity in the brain as a diffusion process over the underlying anatomical pathways (Abdelnour et al., 2014; Liang & Wang, 2017), we show the applicability of GDNs to infer brain structural connectivity from functional networks obtained from the Human Connectome Project-Young Adult (HCP-YA) dataset. We also use GDNs to predict Facebook ties from user co-location data, outperforming relevant baselines.

**Related work.** Network topology inference has a long history in statistics (Dempster, 1972), with noteworthy contributions for probabilistic graphical model selection; see e.g. (Kolaczyk, 2009; Friedman et al., 2008; Drton & Maathuis, 2017). Recent advances were propelled by graph signal processing insights through the lens of signal representation (Dong et al., 2019; Mateos et al., 2019), exploiting models of network diffusion (Daneshmand et al., 2014), or else leveraging cardinal properties of network data such as smoothness (Dong et al., 2016; Kalofolias, 2016) and graph stationarity (Segarra et al., 2017; Pasdeloup et al., 2018). These works formulate (convex) optimization problems one has to solve for different graphs, and can lack robustness to signal model misspecifications. Scalability is an issue for the spectral-based network deconvolution approaches in (Segarra et al., 2017; Feizi et al., 2013), that require computationally-expensive eigendecompositions of the input graph for each problem instance. Moreover, none of these methods advocate a supervised learning paradigm to learn distributions over adjacency matrices. When it comes to this latter objective, deep generative models (Liao et al., 2019; Wang et al., 2018; Li et al., 2018) are typically trained in an unsupervised fashion, with the different goal of sampling from the learnt distribution. Most of these approaches learn node embeddings and are inherently transductive. Recently, so-termed latent graph learning has been shown effective in obtaining better task-driven representations of relational data for machine learning (ML) applications (Wang et al., 2019; Kazi et al., 2020; Veličković et al., 2020), or to learn interactions among coupled dynamical systems (Kipf et al., 2018).

**Summary of contributions.** We introduce GDNs, a supervised learning model capable of recovering latent graph structure from observations of its convolutional mixtures, i.e., related graphs containing spurious, indirect relationships. Our experiments on synthetic and real datasets demonstrate the effectiveness of GDNs for said task. They also showcase the model's versatility to incorporate domain-specific topology information about the sought graphs. On synthetic data, GDNs outperform comparable methods on link prediction and edge-weight regression tasks across different random-graph ensembles, while incurring a markedly lower (post-training) computational cost and inference time. GDNs are inductive and learnt models transfer across graphs of different sizes. We verify they exhibit minimal performance degradation even when tested on graphs $60\times$ larger. Finally, using GDNs we propose a novel ML pipeline to learn whole brain structural connectivity (SC) from functional connectivity (FC), a challenging and timely problem in network neuroscience. Results on the high-quality HCP-YA imaging dataset show that GDN performs well on specific brain subnetworks that are known to be relatively less correlated with the corresponding FC due to ageing-related effects – a testament to the model's robustness and expressive power. Overall, results here support the promising prospect of using graph representation learning to integrate brain structure and function.

## 2 PROBLEM FORMULATION

In this work we study the following network inverse problem involving undirected and weighted graphs $\mathcal{G}(\mathcal{V}, \mathcal{E})$, where $\mathcal{V} = \{1, \ldots, N\}$ is the set of nodes (henceforth common to all graphs), and $\mathcal{E} \subseteq \mathcal{V} \times \mathcal{V}$ collects the edges. We get to observe a graph with symmetric adjacency matrix $\boldsymbol{A}_O \in \mathbb{R}_+^{N \times N}$, that is related to a latent sparse, graph $\boldsymbol{A}_L \in \mathbb{R}_+^{N \times N}$ of interest via the model

$$\boldsymbol{A}_O = \alpha_0 \boldsymbol{I} + \alpha_1 \boldsymbol{A}_L + \alpha_2 \boldsymbol{A}_L^2 + \ldots = \sum_{i=0}^{\infty} \alpha_i \boldsymbol{A}_L^i, \tag{1}$$

where $\boldsymbol{I}$ denotes the $N \times N$ identity matrix. The analytic graph mapping in (1) is a polynomial in $\boldsymbol{A}_L$ of possibly infinite degree, yet the Cayley-Hamilton theorem asserts it can always be equivalently reparameterized as a polynomial of degree smaller than $N$. Said matrix polynomials $\boldsymbol{H}(\boldsymbol{A}; \boldsymbol{h}) := \sum_{k=0}^{K} h_k \boldsymbol{A}^k$, $K \leq N - 1$, with coefficients $\boldsymbol{h} := [h_0, \ldots, h_K]^\top \in \mathbb{R}^{K+1}$, are known as shift-invariant graph convolutional filters; see e.g., (Ortega et al., 2018; Gama et al., 2020b). Going back to the model (1), we postulate that $\boldsymbol{A}_O = \boldsymbol{H}(\boldsymbol{A}_L, \boldsymbol{h})$ for some filter order $K$ and its associated coefficients $\boldsymbol{h}$, such that we can think of the observed network as generated via a graph convolutional process acting on $\boldsymbol{A}_L$. More pragmatically $\boldsymbol{A}_O$ may correspond to a noisy observation of $\boldsymbol{H}(\boldsymbol{A}_L, \boldsymbol{h})$, and this will be clear from the context when e.g., we estimate $\boldsymbol{A}_O$ from data.

Recovery of the latent graph $\boldsymbol{A}_L$ is a challenging endeavour since we do not know $\boldsymbol{H}(\boldsymbol{A}_L; \boldsymbol{h})$, namely the parameters $K$ or $\boldsymbol{h}$; see Appendix A.1 for issues of model identifiability. Suppose that $\boldsymbol{A}_L$ is a realization drawn from some distribution of sparse graphs, say, for e.g., random geometric graphs or structural brain networks from a homogeneous dataset. Then given independent training samples $\mathcal{T} := \{\boldsymbol{A}_O^{(i)}, \boldsymbol{A}_L^{(i)}\}_{i=1}^T$ adhering to (1), our goal is to learn a judicious parametric mapping that predicts the graph adjacency matrix $\hat{\boldsymbol{A}}_L = \Phi(\boldsymbol{A}_O; \boldsymbol{\Theta})$ by minimizing a loss function

$$L(\boldsymbol{\Theta}) := \frac{1}{T} \sum_{i \in \mathcal{T}} \ell(\boldsymbol{A}_L^{(i)}, \Phi(\boldsymbol{A}_O^{(i)}; \boldsymbol{\Theta})). \tag{2}$$

The loss $\ell$ is chosen to accommodate the task at hand – hinge loss for link prediction or mean-squared/absolute error for the more challenging edge-weight regression problem; see Appendix A.4.

## 3 MOTIVATING APPLICATION DOMAINS

Here we outline several graph inference tasks that can be cast as the network inverse problem (1).

**Latent graph structure identification from diffused signals.** Our initial focus here is on identifying graphs that explain the structure of a class of network diffusion processes. Formally, let $\boldsymbol{x} \in \mathbb{R}^N$ be a graph signal (i.e., a vector of nodal features) supported on a latent graph $\mathcal{G}$ with adjacency $\boldsymbol{A}_L$. Further, let $\boldsymbol{w}$ be a zero-mean white signal with covariance matrix $\boldsymbol{\Sigma}_w = \mathbb{E}[\boldsymbol{w}\boldsymbol{w}^\top] = \boldsymbol{I}$. We say that $\boldsymbol{A}_L$ represents the structure of the signal $\boldsymbol{x}$ if there exists a linear network diffusion process in $\mathcal{G}$ that generates the signal $\boldsymbol{x}$ from $\boldsymbol{w}$, namely $\boldsymbol{x} = \sum_{i=0}^\infty \alpha_i \boldsymbol{A}_L^i \boldsymbol{w} = \boldsymbol{H}(\boldsymbol{A}_L, \boldsymbol{h})\boldsymbol{w}$. This is a fairly common generative model for random network processes (Barrat et al., 2008; DeGroot, 1974). We think of the edges of $\mathcal{G}$ as direct (one-hop) relations between the elements of the signal $\boldsymbol{x}$. The diffusion described by $\boldsymbol{H}(\boldsymbol{A}_L, \boldsymbol{h})$ generates indirect relations. In this context, the latent graph learning problem is to recover a sparse $\boldsymbol{A}_L$ from a set $\mathcal{X} := \{\boldsymbol{x}_i\}_{i=1}^P$ of $P$ samples of $\boldsymbol{x}$ (Segarra et al., 2017). Interestingly, from the model for $\boldsymbol{x}$ it follows that the signal covariance matrix $\boldsymbol{\Sigma}_x = \mathbb{E}\left[\boldsymbol{x}\boldsymbol{x}^\top\right] = \boldsymbol{H}^2$ is also a polynomial in $\boldsymbol{A}_L$ (we used $\boldsymbol{\Sigma}_w = \boldsymbol{I}$, and wrote $\boldsymbol{H} \leftarrow \boldsymbol{H}(\boldsymbol{A}_L, \boldsymbol{h})$ for notational simplicity). The connection with (1) should now be apparent with the identification $\boldsymbol{A}_O = \boldsymbol{\Sigma}_x$. In practice, given the signals in $\mathcal{X}$ one would estimate the covariance matrix, e.g. via the sample covariance $\hat{\boldsymbol{\Sigma}}_x$, and then aim to recover the graph $\boldsymbol{A}_L$ by tackling the aforementioned network inverse problem. In this paper, we propose a fresh learning-based solution using training examples $\mathcal{T} := \{\hat{\boldsymbol{\Sigma}}_x^{(i)}, \boldsymbol{A}_L^{(i)}\}_{i=1}^T$.

**Network deconvolution and denoising.** The network deconvolution problem is to identify a sparse adjacency matrix $\boldsymbol{A}_L$ that encodes direct dependencies, when given an adjacency matrix $\boldsymbol{A}_O$ of indirect relationships. The problem broadens the scope of e.g., signal deconvolution to networks and can be tackled by attempting to invert the mapping $\boldsymbol{A}_O = \boldsymbol{A}_L (\boldsymbol{I} - \boldsymbol{A}_L)^{-1} = \sum_{i=1}^\infty \boldsymbol{A}_L^i$. This solution proposed in (Feizi et al., 2013) assumes a polynomial relationship as in (1), but for the particular case of a single-pole, single-zero graph filter with very specific filter coefficients [cf. (1) with $\alpha_0 = 0$ and $\alpha_i = 1$, $i \geq 1$]. This way, the indirect dependencies observed in $\boldsymbol{A}_O$ arise due to the higher-order convolutive mixture terms $\boldsymbol{A}_L^2 + \boldsymbol{A}_L^3 + \ldots$ superimposed to the direct interactions in $\boldsymbol{A}_L$ we wish to recover. Our idea in this paper is to adopt a more general, data-driven learning approach in assuming that $\boldsymbol{A}_O$ can be written as a polynomial in $\boldsymbol{A}_L$, but being agnostic to the form of the filter. Unlike the problem outlined before, here $\boldsymbol{A}_O$ need not be a covariance matrix. Indeed, $\boldsymbol{A}_O$ could be a corrupted graph we wish to denoise, obtained via an upstream graph learning method. Potential application domains for which supervised data $\mathcal{T} := \{\boldsymbol{A}_O^{(i)}, \boldsymbol{A}_L^{(i)}\}_{i=1}^T$ is available include bioinformatics [infer protein contact structure from mutual information graphs of the covariation of amino acid residues (Feizi et al., 2013)], social and information networks [e.g., learn to sparsify graphs (Spielman & Srivastava, 2011) to unveil the most relevant collaborations in a social network encoding co-authorship information (Segarra et al., 2017)], and epidemiology (such as contact tracing by deconvolving the graphs that model observed disease spread in a population). In Section 5.2 we experiment with social networks and the network neuroscience problem described next.

**Inferring structural brain networks from functional MRI (fMRI) signals.** Brain connectomes encompass networks of brain regions connected by (statistical) functional associations (FC) or by anatomical white matter fiber pathways (SC). The latter can be extracted from time-consuming tractography algorithms applied to diffusion MRI (dMRI), which are particularly fraught due to quality issues in the data (Yeh et al., 2021). FC represents pairwise correlation structure between blood-

oxygen-level-dependent (BOLD) signals measured by fMRI. Deciphering the relationship between SC and FC is a very active area of research (Abdelnour et al., 2014; Honey et al., 2009) and also relevant in studying neurological disorders, since it is known to vary with respect to healthy subjects in pathological contexts (Gu et al., 2021). Traditional approaches go all the way from correlation studies (Greicius et al., 2008) to large-scale simulations of nonlinear cortical activity models (Honey et al., 2009). More aligned with the problem addressed here, recent studies have shown that linear diffusion dynamics can reasonably model the FC-SC coupling (Abdelnour et al., 2014; Surampudi et al., 2018). Using our notation, the findings in (Abdelnour et al., 2014) suggest that the covariance $\boldsymbol{A}_O = \boldsymbol{\Sigma}_x$ of the functional signals (i.e., the FC) is related to the the sparse SC graph $\boldsymbol{A}_L$ via the model in (1). Similarly, Liang & Wang (2017) contend the estimated FC can be represented as a weighted sum of the powers of the SC matrix, consisting of both direct and indirect effects along varying paths. There is evidence that FC links tend to exist where there is no or little structural connection (Damoiseaux & Greicius, 2009), a characteristic naturally captured by (1). These considerations motivate adopting the proposed graph learning method to infer SC patterns from fMRI signals (Section 5.2), a significant problem for several reasons. The ability to collect only FC and get informative estimates of SC open the door to large scale studies, previously constrained by the logistical, cost, and computational resources needed to acquire both modalities.

# 4 GRAPH DECONVOLUTION NETWORK

Here we present the proposed GDN model, a parameterized neural network architecture that we train in a supervised fashion to recover latent graph structure via network deconvolution. In the sequel, we obtain conceptual iterations to tackle an optimization formulation of the network inverse problem (Section 4.1), unroll these iterations to arrive at the GDN model we train using graph data (Section 4.2), and describe architectural customizations to improve performance (Section 4.3).

## 4.1 MOTIVATION VIA ITERATIVE OPTIMIZATION

Going back to the inverse problem of recovering a sparse adjacency matrix $\boldsymbol{A}_L$ from the mixture $\boldsymbol{A}_O$ in (1), if the graph convolutional filter $\boldsymbol{H}(\boldsymbol{A}; \boldsymbol{h})$ were known we could attempt to solve

$$\hat{\boldsymbol{A}}_L \in \underset{\boldsymbol{A} \in \mathcal{A}}{\arg \min} \left\{ \|\boldsymbol{A}\|_1 + \frac{\lambda}{2} \|\boldsymbol{A}_O - \boldsymbol{H}(\boldsymbol{A}; \boldsymbol{h})\|_F^2 \right\}, \tag{3}$$

where the regularization parameter $\lambda > 0$ trades off sparsity for reconstruction error. The convex set $\mathcal{A} := \{\boldsymbol{A} \in \mathbb{R}^{N \times N} \mid \text{diag}(\boldsymbol{A}) = \boldsymbol{0}, A_{ij} = A_{ij} \geq 0, \forall i, j \in \{1, \ldots, N\}\}$ encodes the admissibility constraints on the adjacency matrix of an undirected graph: hollow diagonal, symmetric, with non-negative edge weights. The $\ell_1$ norm encourages sparsity in the solution, being a convex surrogate of the edge-cardinality function that counts the number of non-zero entries in $\boldsymbol{A}$. Since $\boldsymbol{A}_O$ is often a noisy observation or estimate of the polynomial $\boldsymbol{H}(\boldsymbol{A}_L; \boldsymbol{h})$, it is prudent to relax the equality (1) and minimize the squared residual errors instead.

The composite cost in (3) is a weighted sum of a non-smooth function $\|\boldsymbol{A}\|_1$ and a continuously differentiable function $g(\boldsymbol{A}) := \frac{1}{2} \|\boldsymbol{A}_O - \boldsymbol{H}(\boldsymbol{A}; \boldsymbol{h})\|_F^2$. Notice though that $g(\boldsymbol{A})$ is non-convex and its gradient is only locally Lipschitz continuous due to the graph filter $\boldsymbol{H}(\boldsymbol{A}; \boldsymbol{h})$; except when $K = 1$, but the affine case is not interesting since $\boldsymbol{A}_O$ is just a scaled version of $\boldsymbol{A}_L$. By virtue of the polynomial structure of the non-convexity, provably convergent iterations can be derived using e.g., the Bregman proximal gradient method (Bolte et al., 2018) for judiciously chosen kernel generating distance; see also (Zhang & Hong, 2020). But our end goal here is not to solve (3) iteratively, recall we cannot even formulate the problem because $\boldsymbol{H}(\boldsymbol{A}; \boldsymbol{h})$ is unknown. To retain the essence of the problem structure and motivate a parametric model to learn approximate solutions, it suffices to settle with conceptual proximal gradient (PG) iterations ($k$ henceforth denote iterations, $\boldsymbol{A}[0] \in \mathcal{A}$)

$$\boldsymbol{A}[k+1] = \text{ReLU}(\boldsymbol{A}[k] - \tau \nabla g(\boldsymbol{A}[k]) - \tau \boldsymbol{1}\boldsymbol{1}^\top) \quad k = 0, 1, 2, \ldots, \tag{4}$$

where $\tau$ is a step-size parameter in which we have absorbed $\lambda$. These iterations implement a gradient descent step on $g$ followed by the $\ell_1$ norm's proximal operator; for more on PG algorithms see (Parikh & Boyd, 2014). Due to the non-negativity constraints in $\mathcal{A}$, the $\ell_1$ norm's proximal operator takes the form of a $\tau$-shifted ReLU on the off-diagonal entries of its matrix argument. Also, the operator sets $\text{diag}(\boldsymbol{A}[k+1]) = \boldsymbol{0}$. In the next section, we unroll and truncate these iterations to arrive at the trainable GDN parametric model $\Phi(\boldsymbol{A}_O; \boldsymbol{\Theta})$.

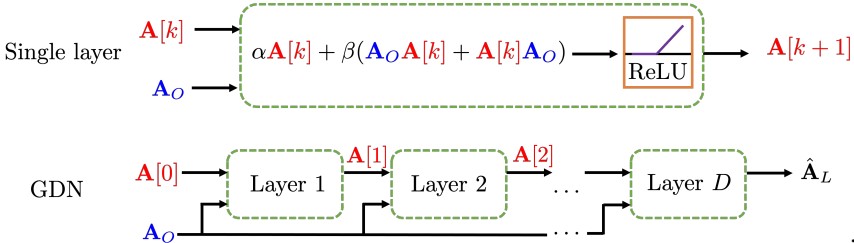

Figure 1: Schematic diagram of the GDN architecture obtained via algorithm unrolling.

## 4.2 LEARNING TO INFER GRAPHS VIA ALGORITHM UNROLLING

The idea of algorithm unrolling can be traced back to the seminal work of (Gregor & LeCun, 2010). In the context of sparse coding, they advocated identifying *iterations* of PG algorithms with *layers* in a deep network of fixed depth that can be trained from examples using backpropagation. One can view this process as effectively truncating the iterations of an asymptotically convergent procedure, to yield a template architecture that learns to approximate solutions with substantial computational savings relative to the optimization algorithm. Beyond parsimonious signal modeling, there has been a surge in popularity of unrolled deep networks for a wide variety of applications; see e.g., (Monga et al., 2021) for a recent tutorial treatment focused on signal and image processing. However, to the best of our knowledge this approach is yet to be explored for latent graph learning.

Building on the algorithm unrolling paradigm, our idea is to design a non-linear, parameterized, feed-forward architecture that can be trained to predict the latent graph $\hat{A}_L = \Phi(A_O; \Theta)$. To this end, we approximate the gradient $\nabla g(A)$ by retaining only linear terms in $A$, and build a deep network by composing layer-wise linear filters and point-wise nonlinearites to capture higher-order interactions in the generative process $H(A; h) := \sum_{k=0}^{K} h_k A^k$. In more detail, we start by simplifying $\nabla g(A)$ (derived in Appendix A.6) by dropping all higher-order terms in $A$, namely

$$\nabla g(A) = -\sum_{k=1}^{K} h_k \sum_{r=0}^{k-1} A^{k-r-1} A_O A^r + \frac{1}{2} \nabla_A \operatorname{Tr} \left[ H^2(A; h) \right]$$
$$\approx -h_1 A_O - h_2 (A_O A + A A_O) + (2h_0 h_2 + h_1^2) A. \tag{5}$$

Notice that $\nabla_A \operatorname{Tr} \left[ H^2(A; h) \right]$ is a polynomial of degree $2K - 1$. Hence, we keep the linear term in $A$ but drop the constant offset that is proportional to $I$, which is inconsequential to adjacency matrix updates with null diagonal. An affine approximation will lead to more benign optimization landscapes when it comes to training the resulting GDN model. All in all, the PG iterations become

$$A[k+1] = \operatorname{ReLU}(\alpha A[k] + \beta(A_O A[k] + A[k] A_O) + \gamma A_O - \tau \mathbf{1}\mathbf{1}^\top), \tag{6}$$

where $A[0] \in \mathcal{A}$ and we defined $\alpha := (1 - 2\tau h_0 h_2 - \tau h_1^2)$, $\beta := \tau h_2$, and $\gamma := \tau h_1$. The latter parameter triplet encapsulates filter (i.e., mixture) coefficients and the $\lambda$-dependent algorithm step-size, all of which are unknown in practice.

The GDN architecture is thus obtained by unrolling the algorithm (6) into a deep neural network; see Figure 1. This entails mapping each individual iteration into a layer and stacking a prescribed number $D$ of layers together to form $\Phi(A_O; \Theta)$. The unknown filter coefficients are treated as learnable parameters $\Theta := \{\alpha, \beta, \gamma, \tau\}$, which are shared across layers as in recurrent neural networks (RNNs). The reduced number of parameters relative to most typical neural networks is a characteristic of unrolled networks (Monga et al., 2021). In the next section, we will explore a few customizations to the architecture in order to broaden the model's expressive power. Given a training set $\mathcal{T} = \{A_O^{(i)}, A_L^{(i)}\}_{i=1}^{T}$, learning is accomplished by using mini-batch stochastic gradient descent to minimize the task-dependent loss function $L(\Theta)$ in (2). We adopt a hinge loss for link prediction and mean-squared/absolute error for the edge-weight regression task. For link prediction, we also learn a threshold $t \in \mathbb{R}_+$ to binarize the estimated edge weights and declare presence or absence of edges; see Appendix A.4 for all training-related details including those concerning loss functions.

The iterative refinement principle of optimization algorithms naturally carries over to our GDN model during inference. Indeed, we start with an initial estimate $A[0] \in \mathcal{A}$ and use a cascade of

$D$ linear filters and point-wise non-linearities to refine it to an output $\hat{\boldsymbol{A}}_L = \Phi(\boldsymbol{A}_O; \boldsymbol{\Theta})$. Matrix $\boldsymbol{A}[0]$ is a hyperparameter we can select to incorporate prior information on the sought latent graph, or it could be learned; see Section 4.3. The input graph $\boldsymbol{A}_O$ to deconvolve is directly fed to all layers as in a residual neural network, and its role is also noteworthy. First, the constant matrix $\gamma \boldsymbol{A}_O - \tau \mathbf{1} \mathbf{1}^\top$ defines non-uniform soft thresholds to effectively sparsify the filter output per layer. Second, one can interpret $\alpha \boldsymbol{A} + \beta(\boldsymbol{A}_O \boldsymbol{A} + \boldsymbol{A} \boldsymbol{A}_O)$ as a first-order graph filter defined on $\boldsymbol{A}_O$, which is used to process $\boldsymbol{A}$ – here viewed as a (graph) signal with $N$ features per node to invoke this graph signal processing insight. In its simplest rendition, the GDN architecture brings together elements of RNNs, ResNets and graph convolutional networks (GCNs) (Kipf & Welling, 2017) .

### 4.3 GDN ARCHITECTURE ADAPTATIONS

Here we outline several customizations and enhancements to the vainilla GDN architecture of the previous section, which we have empirically found to improve graph learning performance.

**Incorporating prior information via algorithm initialization.** By viewing our method as an iterative refinement of an initial graph $\boldsymbol{A}[0]$, one can think of $\boldsymbol{A}[0]$ as a best initial guess, or *prior*, over $\boldsymbol{A}_L$. A simple strategy to incorporate prior information about some edge $(i, j)$, encoded in $A_{ij}$ that we view as a random variable, would be to set $A[0]_{ij} = \mathbb{E}[A_{ij}]$. This technique is adopted when training on the HCP-YA dataset in Section 5.2, by taking the prior $\boldsymbol{A}[0]$ to be the sample mean of all latent (i.e., SC) graphs in the training set. This encodes our prior knowledge that there are strong similarities in the structure of the human brain across the population of healthy young adults. When $\boldsymbol{A}_L$ is expected to be reasonably sparse, we can set $\boldsymbol{A}[0] = \boldsymbol{0}$, which is effective as we show in Table 1. Recalling the connections drawn between GDNs and RNNs, then the prior $\boldsymbol{A}[0]$ plays a similar role to the initial RNN input and thus it could be learned. In any case, the ability to seamlessly incorporate prior information to the model is an attractive feature of GDNs, and differentiates it from other methods trying to solve the network inverse problem.

**Multi-Input Multi-Output (MIMO) filters.** So far, in each layer we have a single learned filter, which takes an $N \times N$ matrix as input and returns another $N \times N$ matrix at the output. After going through the shifted ReLU nonlinearity, this refined output adjacency matrix is fed to the input to the next layer; a process that repeats $D$ times. More generally, we can allow for multiple input channels (i.e., a tensor), as well as multiple channels at the output, by using the familiar convolutional neural network (CNN) methodology. This way, each output channel has its own filter parameters associated with every input channel. The $j$-th output channel applies its linear filters to all input channels, aggregating the results with a reduction operation (e.g., mean or sum), and applies a point-wise nonlinearity (here a shifted ReLU) to the output. This allows the GDN model to learn many different filters, providing richer learned representations. We denote this more expressive architecture as GDN-share-$C$, emphasizing the MIMO filter with $C$ input and output channels and whose parameters are shared across layers. Full details of MIMO filters are given in Appendix A.5.

**Decoupling layer parameters.** Thus far, we have respected the parameter sharing constraint imposed by the unrolled PG iterations. We now allow each layer to learn a decoupled MIMO filter, with its own set of parameters mapping from $C_{in}^k$ input channels to $C_{out}^k$ output channels. As the notation suggests, $C_{in}^k$ and $C_{out}^k$ need not be equal. By decoupling the layer structure, we allow GDNs to compose different learned filters to create more abstract features (as with CNNs or GCNs). Accordingly, it opens up the architectural design space to broader exploration, e.g., wider layers early and skinnier layers at the end. Exploring this architectural space is beyond the scope of this paper and is left as future work. In subsequent experiments, the GDN model for which intermediate layers $k \in \{2, \dots, D-1\}$ have $C = C_{in}^k = C_{out}^k$, i.e., a flat architecture, is denoted as GDN-$C$.

## 5 EXPERIMENTS

We present experiments on link prediction and edge-weight regression tasks using synthetic data (Section 5.1), as well as real HCP-YA neuroimgaging and social network data (Section 5.2). In the link-prediction task, performance is evaluated using error $:= \frac{\text{incorrectly predicted edges}}{\text{total possible edges}}$. For regression, we adopt the mean-squared-error (MSE) or mean-absolute-error (MAE) as figures of merit. In the synthetic data experiments we consider three test cases whereby the latent graphs are respectively drawn from ensembles of Erdős-Rényi (ER), random geometric (RG), and Barabási-Albert

Table 1: Mean and standard error of the test performance across both tasks (Top: link-prediction, Bottom: edge-weight regression) on each graph domain. Bold denotes best performance.

| | Models | RG | ER | BA | SC |
|---|---|---|---|---|---|
| **Error (%)** | GDN-8 | $\mathbf{4.6}_{\pm 4.5\text{e-}1}$ | $41.9_{\pm 1.1\text{e-}1}$ | $\mathbf{27.5}_{\pm 1.0\text{e-}3}$ | $\mathbf{8.9}_{\pm 1.7\text{e-}2}$ |
| | GDN-share-8 | $5.5_{\pm 2.4\text{e-}1}$ | $\mathbf{40.8}_{\pm 1.0\text{e-}2}$ | $27.6_{\pm 8.0\text{e-}4}$ | $9.4_{\pm 2.1\text{e-}1}$ |
| | GLASSO | $8.8_{\pm 6.5\text{e-}2}$ | $43.2_{\pm 1.2\text{e-}2}$ | $34.9_{\pm 9.8\text{e-}3}$ | $20.0_{\pm 3.8\text{e-}2}$ |
| | ND | $9.4_{\pm 3.1\text{e-}1}$ | $43.9_{\pm 1.4\text{e-}2}$ | $34.1_{\pm 8.2\text{e-}3}$ | $21.3_{\pm 9.4\text{e-}2}$ |
| | SpecTemp | $11.1_{\pm 3.2\text{e-}1}$ | $44.4_{\pm 6.6\text{e-}2}$ | $30.2_{\pm 1.8\text{e-}1}$ | $30.0_{\pm 1.3\text{e-}1}$ |
| | LSOpt | $24.2_{\pm 4.8\text{e-}0}$ | $42.5_{\pm 2.8\text{e-}1}$ | $28.0_{\pm 2.0\text{e-}1}$ | $31.53_{\pm 5.8\text{e-}3}$ |
| | Threshold | $12.0_{\pm 1.8\text{e-}1}$ | $42.9_{\pm 8.3\text{e-}1}$ | $32.3_{\pm 1.0\text{e-}0}$ | $21.7_{\pm 2.1\text{e-}1}$ |
| **MSE** | GDN-8 | $\mathbf{4.2}\text{e-}2_{\pm 4.3\text{e-}3}$ | $2.3\text{e-}1_{\pm 2.2\text{e-}3}$ | $\mathbf{1.8}\text{e-}1_{\pm 2.4\text{e-}3}$ | $\mathbf{5.3}\text{e-}3_{\pm 6.7\text{e-}5}$ |
| | GDN-share-8 | $6.0\text{e-}2_{\pm 2.4\text{e-}1}$ | $\mathbf{2.3}\text{e-}1_{\pm 2.1\text{e-}3}$ | $2.7\text{e-}1_{\pm 1.6\text{e-}2}$ | $6.5\text{e-}3_{\pm 4.0\text{e-}5}$ |
| | GLASSO | $2.0\text{e-}1_{\pm 2.6\text{e-}3}$ | $2.8\text{e-}1_{\pm 2.0\text{e-}2}$ | $2.6\text{e-}1_{\pm 1.6\text{e-}2}$ | $4.4\text{e-}2_{\pm 3.3\text{e-}5}$ |
| | ND | $1.8\text{e-}1_{\pm 1.5\text{e-}3}$ | $2.4\text{e-}1_{\pm 5.0\text{e-}4}$ | $2.2\text{e-}1_{\pm 1.0\text{e-}3}$ | $5.6\text{e-}2_{\pm 6.8\text{e-}5}$ |
| | SpecTemp | $5.1\text{e-}2_{\pm 3.3\text{e-}5}$ | $5.3\text{e-}1_{\pm 8.9\text{e-}5}$ | $3.3\text{e-}1_{\pm 1.8\text{e-}5}$ | $1.5\text{e-}1_{\pm 4.2\text{e-}3}$ |
| | LSOpt | $9.9\text{e-}2_{\pm 1.7\text{e-}1}$ | $2.5\text{e-}1_{\pm 1.5\text{e-}3}$ | $2.0\text{e-}1_{\pm 2.5\text{e-}3}$ | $6.1\text{e-}0_{\pm 5.8\text{e-}4}$ |

(BA) random graph models. We study an additional scenario where we use SCs from HCP-YA (referred to as the 'pseudo-synthetic' case because the latent graphs are real structural brain networks). We compare GDNs against several relevant baselines: Network Deconvolution (ND) which uses a spectral approach to directly invert a very specific convolutive mixture (Feizi et al., 2013); Spectral Templates (SpecTemp) that advocates a convex optimization approach to recover sparse graphs from noisy estimates of $\boldsymbol{A}_O$'s eigenvectors (Segarra et al., 2017); Graphical LASSO (GLASSO), a regularized MLE of the precision matrix for Gaussian graphical model selection (Friedman et al., 2008); least-squares fitting of $\boldsymbol{h}$ followed by non-convex optimization to solve (3) (LSOpt); and Hard Thresholding (Threshold) to assess how well a simple cutoff rule can perform. Unless otherwise stated, in all the results that follow we use GDN(-share) models with $D = 8$ layers and train using the Adam optimizer (Kingma & Ba, 2015) with learning rate of 0.01 and batch size of 200.

## 5.1 LATENT GRAPH STRUCTURE IDENTIFICATION FROM DIFFUSED SIGNALS

A set of latent graphs are either sampled from RG, ER, or the BA model, or taken as the SCs from the HCP-YA dataset. In an attempt to make the latent graphs somewhat comparable, across all models we let $N = 68$ (as constrained by the regions of interest in the adopted brain atlas), we force connectivity, and edge sparsity levels in the range $[0.5, 0.6]$ when feasible (these are also typical SC values). To generate each observation $\boldsymbol{A}_O^{(i)}$, we simulated $P = 50$ standard Normal, white signals diffused over $\boldsymbol{A}_L^{(i)}$; from which we form the sample covariance $\hat{\boldsymbol{\Sigma}}_x$ as in Section 3. We let $K = 2$, and sample the filter coefficients $\boldsymbol{h} \in \mathbb{R}^3$ in $\boldsymbol{H}(\boldsymbol{A}_L; \boldsymbol{h})$ uniformly from the unit sphere. To examine robustness to the realizations of $\boldsymbol{h}$, we repeat this data generation process three times (resampling the filter coefficients). We thus create three different datasets for each graph domain (12 in total). For the sizes of the training/validation/test splits, the pseudo-synthetic domain uses 913/50/100 and the synthetic domains use 913/500/500. All models on synthetics are trained using $\boldsymbol{A}[0] = \boldsymbol{0}$, while models on the SCs take their prior as the edge-wise mean across all SCs in the training split.

Table 1 tabulates the results for synthetic and pseudo-synthetic experiments. For graph models that exhibit localized connectivity patterns (RG and SC), GDNs significantly outperform the baselines on both tasks. For the SC test case, GDN (GDN-share) reduces error relative to the mean prior by $27.48 \pm 1.73\%$ ($23.02 \pm 1.73\%$) and MSE by $37.34 \pm 0.79\%$ ($23.23 \pm 0.48\%$). Both GDN architectures show the ability to learn such local patterns, with the extra representational power of GDNs (over GDN-share) providing an additional boost in performance. All models struggle on BA and ER with GDNs showing better performance even for these cases.

**Size generalization: Deploying on larger graphs.** Unlike CNNs and GNNs, GDNs learn the parameters of graph convolutions for the processing of graphs, not the signals supported on them. GDNs are inductive and allow us to deploy the learnt model on larger graph size domains. To test

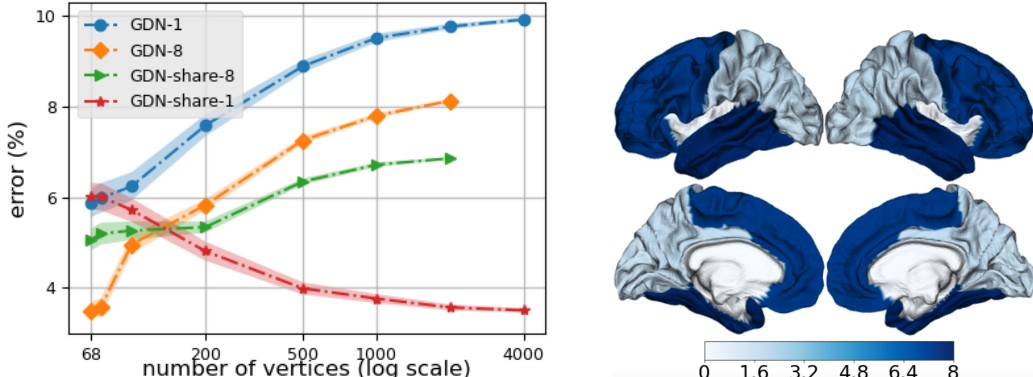

Figure 2: Left: After training on RG graphs of size $N = 68$, GDNs are capable of maintaining performance on RG graphs orders of magnitude larger. Missing data at $N = 4000$ corresponds to overwhelmed our memory resources. The simplest model GDN-share-1 improves performance with increasing graph size. Right: Reduction in MAE (%) over mean prior of SCs for different lobes. Most significant improvements are concentrated in temporal and frontal lobes.

the extent to which GDNs generalize when $N$ grows, we train four GDN models (GDN-share-1, GDN-share-8, GDN-1, GDN-8) on RG graphs with size $N = 68$, and tested them on RG graphs of size $N = [68, 75, 100, 200, 500, 1000, 2000, 4000]$, with 200 graphs of each size. As graph sizes increase, we require more samples in the estimation of the sample covariance to maintain a constant signal-to-noise ratio. To simplify the experiment and its interpretation, we disregard estimation and directly use the ensemble covariance $\boldsymbol{A}_O \equiv \boldsymbol{\Sigma}_x$ as observation. As before, we take a training/validation split of $913/500$. Figure 2 shows all GDN models effectively generalize to graphs orders of magnitude larger than they were trained on, giving up only modest levels of performance as size increases. For $N = 4000$, all but one of GDN models performed better than any baseline did on the original 68 node graphs. The GDN-share models maintain their top performance, with GDN-share-1 even *further reducing error* as the domain gets larger. This suggests that the GDNs without parameter sharing may be using their extra representational power to pick up on finite-size effects, which may disappear as $N$ increases. The shared layer constraint in GDN-share models acts as regularization: we avoid over-fitting on a given size domain to better generalize to larger graphs.

**Ablation studies.** The choice of prior can influence model performance, as well as reduce training time and the number of parameters needed. When run on stochastic block model (SBM) graphs with $N = 21$ nodes and 3 equally-sized communities (within block connection probability of $0.6$, and $0.1$ across blocks), for the link-prediction task GDNs attain an error of $16.8 \pm 2.7e\text{-}2\%$, $16.0 \pm 2.1e\text{-}2\%$, $14.5 \pm 1.0e\text{-}2\%$, $14.3 \pm 8.8e\text{-}2\%$ using a zeros, ones, block diagonal, and learned prior, respectively. The performance improves when GDNs are given an informative prior (here a block diagonal matrix matching the graph communities), with further gains when GDNs are allowed to learn $\boldsymbol{A}[0]$.

We also study the effect of gradient truncation. To derive GDNs we approximate the gradient $\nabla g(\boldsymbol{A})$ by dropping all higher-order terms in $\boldsymbol{A}$ ($K = 1$). The case of $K = 0$ corresponds to further dropping the terms linear in $\boldsymbol{A}$, leading to PG iterations $\boldsymbol{A}[k+1] = \text{ReLU}(\boldsymbol{A}[k] + \gamma \boldsymbol{A}_O - \tau \mathbf{1}\mathbf{1}^\top)$ [cf. (6)]. We run this simplified model with $D = 8$ layers and 8 channels per layer on the same RG graphs presented in Table 1. Lacking the linear term that facilitates information aggregation in the graph, the model is not expressive enough and yields a higher error (MSE) of $25.72 \pm 1.3e\text{-}2\%$ ($1.7e\text{-}1 \pm 4.7e\text{-}4$) for the link-prediction (edge weight regression) task. Models with $K \geq 2$ result in unstable training, which motivates our choice of $K = 1$ in GDNs.

## 5.2 REAL DATA

**HCP-YA neuroimaging dataset.** HCP represents a unifying paradigm to acquire high quality neuroimaging data across studies that enabled unprecedented quality of macro-scale human brain connectomes for analyses in different domains (Glasser et al., 2016). We use the dMRI and resting state

fMRI data from the HCP-YA dataset (Van Essen et al., 2013), consisting of 1200 healthy, young adults (ages: 22-36 years). The SC and FC are projected on a common brain atlas, which is a grouping of cortical structures in the brain to distinct regions. We interpret these regions as nodes in a brain graph. For our experiments, we use the standard Desikan-Killiany atlas (Desikan et al., 2006) with $N = 68$ cortical brain regions. The SC-FC coupling on this dataset is known to be the strongest in occipital lobe and vary with age, sex and cognitive health in other subnetworks (Gu et al., 2021). Under the consideration of the variability in SC-FC coupling across the brain regions, we further group the cortical regions into 4 larger regions called 'lobes': frontal, parietal, temporal, and occipital (the locations of these lobes in the brain are included in Fig. 4 in Appendix A.7). We aim to predict SC, derived from dMRI, using FC, constructed using BOLD signals acquired with fMRI.

From this data, we extracted a dataset of 1063 FC-SC pairs, $\mathcal{T} = \{\boldsymbol{FC}^{(i)}, \boldsymbol{SC}^{(i)}\}_{i=1}^{1063}$ and use a training/validation/test split of 913/50/100. Taking the prior $\boldsymbol{A}[0]$ as the edgewise mean over all SCs in the training split $\mathcal{T}_{train}$: $A[0]_{ij} = \text{mean}\ \{SC_{i,j}^{(1)}, \ldots, SC_{i,j}^{(913)}\}$ and using it directly as a predictor on the test set (tuning a threshold with validation data), we achieve strong performance on link-prediction and edge-weight regression tasks on the whole brain (error = $12.25\%$, MAE = $0.0615$). At the lobe level, the prior achieves relatively higher accuracy in occipital (error = $98.71\%$, MAE = $0.056$) and parietal (error = $93.63\%$ MAE = $0.065$) lobes as compared to temporal (error = $88.76\%$, MAE = $0.05$) and frontal (error = $89.55\%$, MAE = $0.059$) lobes; behavior which is unsurprising as SC in temporal and frontal lobes are affected by ageing and gender related variability in the dataset (Zimmermann et al., 2016). GDNs reduced MAE by $7.62\%$, $7.07\%$, $1.58\%$, and $1.29\%$ in the temporal, frontal, parietal, and occipital networks respectively and $7.95\%$ over the entire brain network, all relative to the already strong mean prior. The four lobe reductions are visualized in Figure 2. Clearly, there was smaller room for improvement in performance over occipital and frontal lobes. We observed the most significant gains over temporal and frontal lobes.

In summary, our pseudo-synthetic experiments in Section 5.1 show that SCs are amenable to learning with GDNs when the SC-FC relationship satisfies (1), a reasonable model given the findings of (Abdelnour et al., 2014). In general, SC-FC coupling can vary widely across both the population and within the sub-regions of an individuals brain for healthy subjects and in pathological contexts. Therefore, our results on HCP-YA dataset could potentially serve as baselines that characterize healthy subjects, and expanding our work to study the deviations in findings on similar data in a pathological context would be of future interest. When trained on the HCP-YA dataset, the GDN model exhibits robust performance over such regions with high variability in SC-FC coupling.

**Friendship recommendation from physical co-location networks.** Here we use GDNs to predict Facebook ties given human co-location data. GDNs are well suited for this deconvolution problem since one can view friendships as direct ties, whereas co-location edges include indirect relationships due to casual encounters in addition to face-to-face contacts with friends. A trained model could then be useful to augment friendship recommendation engines given co-location (behavioral) data. The Thiers13 dataset (Génois & Barrat, 2018) monitored high school students, recording (i) physical co-location interactions with wearable sensors over 5 days; and (ii) social network information via survey. From this we construct a dataset $\mathcal{T}$ of graph pairs, each with $N = 120$ nodes, where $\boldsymbol{A}_O$ are co-location networks, i.e., weighted graphs where the weight of edge $(i, j)$ represents the number of times student $i$ and $j$ came into physical proximity, and $\boldsymbol{A}_L$ is the Facebook subgraph between the same students; further details are in Appendix A.7. We trained a GDN-11, without MIMO filters, with learned $\boldsymbol{A}[0]$ using a training/validation/test split of $5000/1000/1000$. We achieved a test error of $8.9 \pm 1.5e\text{-}2\%$, a $12.98\%$ reduction over next best performing baseline (results in Appendix A.7).

## 6 CONCLUSIONS

In this work we proposed the GDN, an inductive model capable of recovering latent graph structure from observations of its convolutional mixtures. By minimizing a task-dependent loss function, GDNs learn filters to refine initial estimates of the sought latent graphs layer by layer. The unrolled architecture can seamlessly integrate domain-specific prior information about the unknown graph distribution. Moreover, because GDNs: (i) are differentiable functions with respect to their parameters as well as their graph input; and (ii) offer explicit control on complexity (leading to fast inference times); one can envision GDNs as valuable components in larger (even online) end-to-end graph representation learning systems. This way, while our focus here has been exclusively on network topology identification, the impact of GDNs can permeate to broader graph inference tasks.

**Reproducibility statement.** Code for running these experiments has been included in a zipped directory with the submission and contains instructions for configuring a system to run experiments presented above. When randomness is involved, as is the case when constructing the synthetic datasets, sampling white signals for diffusion, constructing a random split of the HCP-YA data, or initializing the parameters of our model before training, we use a consistent and clearly defined random seed, allowing others to reproduce the results presented. For the derivation of the model, we provide further details in Appendix A.6 to supplement those shown in the main paper body (Section 4.1). In Appendix A.7 we refer the readers to the HCP website, where one can download the HCP-YA data, as well as references to the processing pipelines used to construct the FCs and SCs used in this paper. The processed brain data was too large to include with the code, but is available upon request.

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

# A APPENDIX

## A.1 MODEL IDENTIFIABILITY

Without any constraints on $\boldsymbol{h}$ and $\boldsymbol{A}_L$, the problem of recovering $\boldsymbol{A}_L$ from $\boldsymbol{A}_O = \boldsymbol{H}(\boldsymbol{A}_L; \boldsymbol{h})$ as in (1) is clearly non-identifiable. Indeed, if the desired solution is $\boldsymbol{A}_L$ (with associated polynomial coefficients $\boldsymbol{h}$), there is always at least another solution $\boldsymbol{A}_O$ corresponding to the identity polynomial mapping. This is why adding structural constraints like sparsity on $\boldsymbol{A}_L$ will aid model identifiability, especially when devoid of training examples.

It is worth mentioning that (1) implies the eigenvectors of $\boldsymbol{A}_L$ and $\boldsymbol{A}_O$ coincide. So the eigenvectors of the sought latent graph are given once we observe $\boldsymbol{A}_O$, what is left to determine are the eigenvalues. We have in practice observed that for several families of sparse, weighted graphs, the eigenvector information along with the constraint $\boldsymbol{A}_L \in \mathcal{A}$ are sufficient to uniquely specify the graph. Interestingly, this implies that many random weighted graphs can be uniquely determined from their eigenvectors. This strong uniqueness result does not render our problem vacuous, since seldomly in practice one gets to observe $\boldsymbol{A}_O$ (and hence its eigenvectors) error free.

If one were to formally study identifiability of (1) (say under some structural assumptions on $\boldsymbol{A}_L$ and/or the polynomial mapping), then one has to recognize the problem suffers from an inherent scaling ambiguity. Indeed, if given $\boldsymbol{A}_O = \boldsymbol{H}(\boldsymbol{A}_L; \boldsymbol{h})$ which means the pair $\boldsymbol{A}_L$ and $\boldsymbol{h} = [h_0, h_1, \ldots, h_K]^\top$ is a solution, then for any positive scalar $\alpha$ one has that $\alpha \boldsymbol{A}_L$ and $[h_0, h_1/\alpha, \ldots, h_K/(\alpha^K)]^\top$ is another solution. Accordingly, uniqueness claims can only be meaningful modulo this unavoidable scaling ambiguity. But this ambiguity is lifted once we tackle the problem in a supervised learning fashion – our approach in this paper. The training samples in $\mathcal{T} := \{\boldsymbol{A}_O^{(i)}, \boldsymbol{A}_L^{(i)}\}_{i=1}^T$ fix the scaling, and accordingly the GDN can learn the mechanism or mapping of interest $\boldsymbol{A}_O \mapsto \boldsymbol{A}_L$. Hence, an attractive feature of the GDN approach is that by using data, some of the inherent ambiguities in (1) are naturally overcome. In particular, the SpecTemp approach in (Segarra et al., 2017) relies on convex optimization and suffers from this scaling ambiguity, so it requires an extra (rather arbitrary) constraint to fix the scale. The network deconvolution approach in (Feizi et al., 2013) relies on a fixed, known polynomial mapping, and while it does not suffer from these ambiguities it is limited in the graph convolutions it can model.

All in all, the inverse problem associated to (1) is just our starting point to motivate a trainable parametrized architecture $\hat{\boldsymbol{A}}_L = \Phi(\boldsymbol{A}_O; \boldsymbol{\Theta})$, that introduces valuable inductive biases to generate graph predictions. The problem we end up solving is different (recall the formal statement in Section 2) because we rely on supervision using graph examples, thus rendering many of these challenging uniqueness questions less relevant.

## A.2 GRAPH CONVOLUTIONAL MODEL IN CONTEXT

To further elaborate on the relevance and breadth of applicability of the graph convolutional (or network diffusion) signal model $\boldsymbol{x} = \boldsymbol{H}(\boldsymbol{A}_L; \boldsymbol{h})\boldsymbol{w}$, we would like to elucidate connections with related work for graph structure identification. Note that while we used the diffusion-based generative model for our derivations in Section 3, we do not need it as an actual mechanistic process. Indeed, like in (1) the only thing we ask is for the data covariance $\boldsymbol{A}_O = \boldsymbol{\Sigma}_x$ to be some analytic function of the latent graph $\boldsymbol{A}_L$. This is not extraneous to workhorse statistical methods for topology inference, which (implicitly) make specific choices for these mappings, e.g. (i) correlation networks (Kolaczyk, 2009, Ch. 7) rely on the identity mapping $\boldsymbol{\Sigma}_x = \boldsymbol{A}_L$; (ii) Gaussian graphical model selection methods, such as graphical lasso in (Yuan & Lin, 2007; Friedman et al., 2008), adopt $\boldsymbol{\Sigma}_x = \boldsymbol{A}_L^{-1}$; and (iii) undirected structural equation models $\boldsymbol{x} = \boldsymbol{A}_L \boldsymbol{x} + \boldsymbol{w}$ which implies $\boldsymbol{\Sigma}_x = (\boldsymbol{I} - \boldsymbol{A}_L)^{-2}$ (Mateos et al., 2019). Accordingly, these models are subsumed by the general framework we put forth here.

## A.3 INCORPORATING PRIOR INFORMATION

In Section 4.3 we introduce the concept of using prior information in the training of GDNs. We do so by encoding information we may have about the unknown latent graph $\boldsymbol{A}_L$ into $\boldsymbol{A}[O]$, the starting matrix which GDNs iteratively refine. If the $A_L$'s are repeated instances of a graph with fixed nodes,

as is the case with the SCs with respect to the $68$ fixed brain regions, a simple strategy to incorporate prior information about some edge $\mathbf{A}_{L_{i,j}}$, now viewed as a random variable, would be $\boldsymbol{A}_{0_{i,j}} \leftarrow \mathbb{E}[\mathbf{A}_{L_{i,j}}]$. But there is more that can be done. We also can estimate the variance $\mathrm{Var}(\mathbf{A}_{L_{i,j}})$, and use it during the training of a GDN, for example taking $\boldsymbol{A}_{0_{i,j}} \leftarrow \mathcal{N}(\mathbb{E}[\mathbf{A}_{L_{i,j}}], \mathrm{Var}(\mathbf{A}_{L_{i,j}}))$, or even simpler, using a resampling technique and taking $\boldsymbol{A}_{0_{i,j}}$ to be a random sample in the training set. By doing so, we force the GDN to take into account the distribution and uncertainty in the data, possibly leading to richer learned representations and better performance. It also would act as a form of regularization, not allowing the model to converge on the naive solution of outputting the simple expectation prior, a likely local minimum in training space.

## A.4 TRAINING

Training of the GDN model will be performed using stochastic (mini-batch) gradient descent to minimize a task-dependent loss function $L(\boldsymbol{\Theta})$ as in (2). The loss is defined either as (i) the edgewise squared/absolute error between the predicted graph and the true graph for regression tasks, or (ii) a hinge loss with parameter $\gamma \geq 0$, both averaged over a training set $\mathcal{T} := \{\boldsymbol{A}_O^{(i)}, \boldsymbol{A}_L^{(i)}\}_{i=1}^T$, namely

$$\ell(\boldsymbol{A}_L^{(i)}, \Phi(\boldsymbol{A}_O^{(i)}; \boldsymbol{\Theta}))_{\text{hinge}} := \sum_{i,j} \begin{cases} (\Phi(\boldsymbol{A}_O^{(i)}; \boldsymbol{\Theta})_{i,j} - \gamma)^+ & \boldsymbol{A}_{L_{i,j}} = 0 \\ (-\Phi(\boldsymbol{A}_O^{(i)}; \boldsymbol{\Theta})_{i,j} + 1 - \gamma)^+ & \boldsymbol{A}_{L_{i,j}} > 0 \end{cases},$$

$$\ell(\boldsymbol{A}_L^{(i)}, \Phi(\boldsymbol{A}_O^{(i)}; \boldsymbol{\Theta}))_{\text{mse}} := \frac{1}{2} \left\| \boldsymbol{A}_L^{(i)} - \Phi(\boldsymbol{A}_O^{(i)}; \boldsymbol{\Theta}) \right\|_2^2,$$

$$\ell(\boldsymbol{A}_L^{(i)}, \Phi(\boldsymbol{A}_O^{(i)}; \boldsymbol{\Theta}))_{\text{mae}} := \left\| \boldsymbol{A}_L^{(i)} - \Phi(\boldsymbol{A}_O^{(i)}; \boldsymbol{\Theta}) \right\|_1,$$

$$L(\boldsymbol{\Theta}) := \frac{1}{T} \sum_{i \in \mathcal{T}} \ell_u(\boldsymbol{A}_L^{(i)}, \Phi(\boldsymbol{A}_O^{(i)}; \boldsymbol{\Theta})), \qquad u \in \{\text{hinge, mse, mae}\}.$$

**Link prediction with GDNs and unbiased estimates of generalization.** In the edge-weight regression task, GDNs only use their validation data to determine when training has converged. When performing link-prediction, GDNs have an additional use for this data: to choose the cutoff threshold $t \in \mathbb{R}_+$, determining which raw outputs (which are continuous) should be considered positive edge predictions, *at the end of training*.

We use the training set to learn the parameters (via gradient descent) *and* to tune $t$. During the validation step, when then use this train-set-tuned-$t$ on the validation data, giving an estimate of generalization error. This is then used for early-stopping, determining the best model learned after training, etc. We do not use the validation data to tune $t$ during training. Only after training has completed, do we tune $t$ with validation data. We train a handful of models this way, and the model which produces the best validation score (in this case lowest error) is the tested with the validation-tuned-$t$, thus providing an unbiased estimate of generalization.

## A.5 MIMO MODEL ARCHITECTURE

**MIMO filters.** Formally, the more expressive GDN architecture with MIMO (Multi-Input Multi-Output) filters is constructed as follows. At layer $k$ of the neural network, we take a three-way tensor $\mathbf{A}_k \in \mathbb{R}_+^{C \times N \times N}$ and produce $\mathbf{A}_{k+1} \in \mathbb{R}_+^{C \times N \times N}$, where $C$ is the common number of input and output channels. The assumption of having a common number of input and output channels can be relaxed, as we argue below. By defining multiplication between tensors $\mathbf{T}, \mathbf{B} \in \mathbb{R}^{C \times N \times N}$ as batched matrix multiplication: $[\mathbf{TB}]_{j,:,:} := \mathbf{T}_{j,:,:} \mathbf{B}_{j,:,:}$, and tensor-vector addition and multiplication as $\mathbf{T} + \boldsymbol{v} := \mathbf{T} + [v_1 \mathbf{1}\mathbf{1}^\top; \ldots; v_C \mathbf{1}\mathbf{1}^\top]$ and $[\boldsymbol{v}\mathbf{T}]_{j,:,:} := v_j \mathbf{T}_{j,:,:}$ respectively for $\boldsymbol{v} \in \mathbb{R}^C$, all operations extend naturally.

Using these definitions, the $j$-th output slice of layer $k$ is

$$[\mathbf{A}_{k+1}]_{j,:,:} = \text{ReLU}[\overline{\boldsymbol{\alpha}_{:,j}\mathbf{A}_k + \boldsymbol{\beta}_{:,j}(\mathbf{A}_O \mathbf{A}_k + \mathbf{A}_k \mathbf{A}_O) + \boldsymbol{\gamma}_{:,j}\mathbf{A}_O} - \tau_j \mathbf{1}\mathbf{1}^\top], \tag{7}$$

where $\overline{\cdots}$ represents the mean reduction over the filtered input channels and the parameters are $\boldsymbol{\alpha}, \boldsymbol{\beta}, \boldsymbol{\gamma} \in \mathbb{R}^{C \times C}$, $\boldsymbol{\tau} \in \mathbb{R}_+^C$. We now take $\boldsymbol{\Theta} := \{\boldsymbol{\alpha}, \boldsymbol{\beta}, \boldsymbol{\gamma}, \boldsymbol{\tau}\}$ for a total of $C \times (3C + 1)$ trainable parameters.

We typically have a single prior matrix $A[0]$ and are interested in predicting a single adjacency matrix $\hat{A}_L$. Accordingly, we construct a new tensor prior $\mathbf{A}[0] := [A[0], \ldots, A[0]] \in \mathbb{R}^{C \times N \times N}$ and (arbitrarily) designate the first output channel as our prediction $\hat{A}_L = \Phi(A_O; \Theta) = [\mathbf{A}_{k+1}]_{1,:,:}$.

We can also allow each layer to learn a decoupled MIMO filter, with its own set of parameters mapping from $C_{in}^k$ input channels to $C_{out}^k$ output channels. As the notation suggests, $C_{in}^k$ and $C_{out}^k$ need not be equal. Layer $k$ now has its own set of parameters $\Theta^k = (\alpha^k, \beta^k, \gamma^k, \tau^k)$, where $\alpha^k, \beta^k, \gamma^k \in \mathbb{R}^{C_{out}^k \times C_{in}^k}$ and $\tau^k \in \mathbb{R}_+^{C_{out}^k}$, for a total of $C_{out}^k \times (3C_{in}^k + 1)$ trainable parameters. The tensor operations mapping inputs to outputs remains basically unchanged with respect to (7), except that the filter coefficients will depend on $k$.

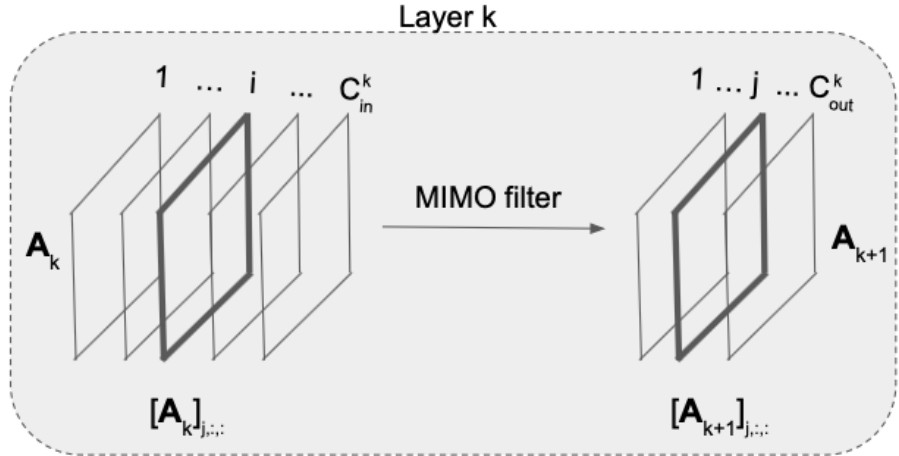

Figure 3: MIMO Filter: Layer $k$ takes a tensor $\mathbf{A}_k \in \mathbb{R}^{C_{in}^k \times N \times N}$ and outputs a tensor $\mathbf{A}_{k+1} \in \mathbb{R}^{C_{out}^k \times N \times N}$. The $i$-th slice $[\mathbf{A}_k]_{i,:,:}$ is called the $i$-th input channel and $[\mathbf{A}_{k+1}]_{j,:,:}$ is called the $j$-th output channel.

Processing at a generic layer $k$ is depicted in Figure 3. Output channel $j$ will use $\alpha_{:,j}^k, \beta_{:,j}^k, \gamma_{:,j}^k \in \mathbb{R}^{C_{in}^k}$ and $\tau_j^k \in \mathbb{R}_+$ to filter all input channels $i \in \{1, \cdots, C_{in}^k\}$, which are collected in the input tensor $\mathbf{A}_k \in \mathbb{R}^{C_{in}^k \times N \times N}$. This produces a tensor of stacked filtered input channels $\in \mathbb{R}^{C_{in}^k \times N \times N}$. After setting the diagonal elements of all matrix slices in this tensor to 0, then perform a mean reduction edgewise (over the first mode/dimension) of this tensor, producing a single $N \times N$ matrix. We then apply two pointwise/elementwise operations on this matrix: (i) subtract $\tau_j^k$ (this would be the 'bias' term in CNNs); and (ii) apply a point-wise nonlinearity (ReLU). This produces an $N \times N$ activation stored in the $j$-th output channel. Doing so for all output channels $j \in \{1, \cdots, C_{out}^k\}$, produces a tensor $\mathbf{A}_{k+1} \in \mathbb{R}^{C_{out}^k \times N \times N}$.

**Layer output normalization and zeroing diagonal.** The steps not shown in the main text are the normalization steps, a practical issue, and the setting of the diagonal elements to be 0, a projection onto the set $\mathcal{A}$ of allowable adjacency matrices. Define $\overline{\mathbf{U}}_{j,:,:} = \overline{\alpha_{:,j}^k \mathbf{A}_k + \beta_{:,j}^k (\mathbf{A}_O \mathbf{A}_k + \mathbf{A}_k \mathbf{A}_O) + \gamma_{:,j}^k \mathbf{A}_O} \in \mathbb{R}^{N \times N}$ as the $j$-th slice in the intermediate tensor $\overline{\mathbf{U}} \in \mathbb{R}^{C_{out}^k \times N \times N}$ used in the filtering of $\mathbf{A}_k$. Normalization is performed by dividing each matrix slice of $\overline{\mathbf{U}}$ by the maximum magnitude element in that respective slice: $\overline{\mathbf{U}}_{\cdot,:,:} / \max |\overline{\mathbf{U}}_{\cdot,:,:}|$.

Multiple normalization metrics were tried on the denominator, including the $99th$ percentile of all values in $\overline{\mathbf{U}}_{\cdot,:,:}$, the Frobenius norm $\|\overline{\mathbf{U}}_{\cdot,:,:}\|_F$, among others. None seemed to work as well as the maximum magnitude element, which has the additional advantage of guaranteeing entries to be in $[0, 1]$ (after ReLU), which matches nicely with: (i) adjacency matrices of unweighted graphs; and (ii) makes it easy to normalize edge weights of a dataset of adjacencies: simply scale them to $[0, 1]$.

In summary, the full procedure to produce $[\mathbf{A}_{k+1}]_{j,:,:}$ is as follows:

$$\overline{\mathbf{U}}_{j,:,:} = \overline{\boldsymbol{\alpha}^k_{:,j}\mathbf{A}_k + \beta^k_{:,j}(\mathbf{A}_O\mathbf{A}_k + \mathbf{A}_k\mathbf{A}_O) + \boldsymbol{\gamma}^k_{j,:})\mathbf{A}_O}$$

$$\overline{\mathbf{U}}_{j,:,:} = \overline{\mathbf{U}}_{j,:,:} \odot (\mathbf{1}\mathbf{1}^\top - \boldsymbol{I}) \qquad\qquad \text{force diagonal elements to 0}$$

$$\overline{\mathbf{U}}_{j,:,:} = \overline{\mathbf{U}}_{j,:,:} / \max(|\overline{\mathbf{U}}_{j,:,:}|) \qquad\qquad \text{normalize entries per slice to be in } [-1, 1]$$

$$[\mathbf{A}_{k+1}]_{j,:,:} = \mathrm{ReLU}(\overline{\mathbf{U}}_{j,:,:} - \tau^j_l)$$

By normalizing in this way, we guarantee the intermediate matrix $\overline{\mathbf{U}}_{j,:,:}$ has entries in $[-1, 1]$ (before the ReLU). This plays two important roles. The first one has to do with training stability and to appreciate this point consider what could happen if no normalization is used. Suppose the entries of $\overline{\mathbf{U}}_{j,:,:}$ are orders of magnitude larger than entries of $\overline{\mathbf{U}}_{l,:,:}$. This can cause the model to push $\tau^k_j >> \tau^k_l$, leading to training instabilities and/or lack of convergence. The second point relates to interpretability of $\tau$. Indeed, the proposed normalization allows us to interpret the learned values $\boldsymbol{\tau}^k \in \mathbb{R}^k_{out,+}$ on a similar scale. All the tresholds must be in $[0, 1]$ because: (i) anything above 1 will make the output all 0; and (ii) we constrain it to be non-negative. In fact we can now plot all $\tau$ values (from all layers) against one another, and using the same scale ($[0, 1]$) interpret if a particular $\tau$ is promoting a lot of sparsity in the output ($\tau$ close to 1) or not ($\tau$ close to 0), by examining its magnitude.

## A.6 GRADIENT USED IN PROXIMAL GRADIENT ITERATIONS

Here we give mathematical details in the calculation of the gradient $\nabla g(\boldsymbol{A})$ of the component function $g(\boldsymbol{A}) := \frac{1}{2}\|\boldsymbol{A}_O - \boldsymbol{H}(\boldsymbol{A}; \boldsymbol{h})\|^2_F$ in the objective function of. Let $\boldsymbol{A}$, $\boldsymbol{A}_O$ be symmetric $N \times N$ matrices and recall the graph filter $\boldsymbol{H}(\boldsymbol{A}) := \sum_{k=0}^K h_k \boldsymbol{A}^k$ (we drop the dependency in $\boldsymbol{h}$ to simply the notation). Then

$$\nabla_{\boldsymbol{A}} \frac{1}{2}\|\boldsymbol{A}_O - \boldsymbol{H}(\boldsymbol{A})\|^2_F = \frac{1}{2}\nabla_{\boldsymbol{A}} \mathrm{Tr}\,(\boldsymbol{A}_O^2 - \boldsymbol{A}_O\boldsymbol{H}(\boldsymbol{A}) - \boldsymbol{H}(\boldsymbol{A})\boldsymbol{A}_O + \boldsymbol{H}^2(\boldsymbol{A}))$$

$$= -\nabla_{\boldsymbol{A}} \mathrm{Tr}\,(\boldsymbol{A}_O\boldsymbol{H}(\boldsymbol{A})) + \frac{1}{2}\nabla_{\boldsymbol{A}} \mathrm{Tr}\,\boldsymbol{H}^2(\boldsymbol{A})$$

$$= -\sum_{k=1}^K h_k \sum_{r=0}^{k-1} \boldsymbol{A}^{k-r-1}\boldsymbol{A}_O\boldsymbol{A}^r + \frac{1}{2}\nabla_{\boldsymbol{A}} \mathrm{Tr}\,\boldsymbol{H}^2(\boldsymbol{A})$$

$$= -\sum_{k=1}^K h_k \sum_{r=0}^{k-1} \boldsymbol{A}^{k-r-1}\boldsymbol{A}_O\boldsymbol{A}^r + \frac{1}{2}\boldsymbol{H}_1(\boldsymbol{A})$$

$$= -[h_1\boldsymbol{A}_O + h_2(\boldsymbol{A}\boldsymbol{A}_O + \boldsymbol{A}_O\boldsymbol{A}) + h_3(\boldsymbol{A}^2\boldsymbol{A}_O + \boldsymbol{A}\boldsymbol{A}_O\boldsymbol{A} + \boldsymbol{A}_O\boldsymbol{A}^2) + \dots]$$

$$+ \frac{1}{2}\boldsymbol{H}_1(\boldsymbol{A}),$$

where in arriving at the second equality we relied on the cyclic property of the trace, and $\boldsymbol{H}_1(\boldsymbol{A})$ is a matrix polynomial of order $2K - 1$.

Note that in the context of the GDN model, powers of $\boldsymbol{A}$ will lead to complex optimization landscapes, and thus unstable training. We thus opt to drop the higher-order terms and work with a first-order approximation of $\nabla g$, namely

$$\nabla g(\boldsymbol{A}) \approx -h_1\boldsymbol{A}_O - h_2(\boldsymbol{A}_O\boldsymbol{A} + \boldsymbol{A}\boldsymbol{A}_O) + (2h_0h_2 + h_1^2)\boldsymbol{A}.$$

## A.7 NOTES ON THE EXPERIMENTAL SETUP

**Synthetic graphs.** For the experiments presented in Table 1, the synthetic graphs of size $N = 68$ are drawn from random graph models with the following parameters

- Random geometric graphs (RG): $d = 2$, $r = 0.56$.
- Erdős-Rényi (ER): $p = .56$.

- Barabási-Albert (BA): $m = 15$

When sampling graphs to construct the datasets, we reject any samples which are not connected or have sparsity outside of a given range. For RG and ER, that range is $[0.5, 0.6]$, while in BA the range is $[0.3, 0.4]$. This is an attempt to make the RG and ER graphs similar to the brain SC graphs, which have an average sparsity of $0.56$, and all SCs are in sparsity range $[0.5, 0.6]$. Due to the sampling procedure of BA, it is not possible to produce graph in this sparsity range, so we lowered the range sightly. We thus take SCs to be an SBM-like ensemble and avoid a repetitive experiment with randomly drawn SBM graphs.

Note that the sparsity ranges define the performance of the most naive of predictors: all ones/zeros. In the RG/BA/SC, an all ones predictor achieves an average error of $44\% = 1-$(average graph sparsity). In the BAs, a naive all zeros predictor achieves $35\% = 1-$(average graph sparsity). This is useful to keep in mind when interpreting the results in Table 1.

**Pseudo-synthetics.** The Pseudo-Synthetic datasets are those in which we diffuse synthetic signals over SCs from the HCP-YA dataset. This is an ideal setting to test the GDN models: we have weighted graphs to perform edge-regression on (the others are unweighted), while having $A_O$'s that are true to our modeling assumptions. Note that SCs have a strong community-like structure, corresponding dominantly to the left and right hemispheres as well as subnetworks which have a high degree of connection, e.g. the Occipital Lobe which has $0.96$ sparsity - almost fully connected - while the full brain network has sparsity of $0.56$.

**Error and MSE of the baselines in Table 1.** The edge weights returned by the baselines can be very small/large in magnitude and perform poorly when used directly in the regression task. We thus also provide a scaling parameter, tuned during the hyperparameter search, which provides approximately an order of magnitude improvement in MSE in GLASSO and halved the MSE in Spectral Templates and Network Deconvolution. In link-prediction, we also tune a hard thresholding parameter on top of each method to clean up noisy outputs from the baseline, only if it improved their performance (it does). For complete details on the baseline implementation, see A.8.

**Size generalization.** Something to note is that we do **not** tune the threshold (found during training on the small $N = 68$ graphs) on the larger graphs. We go straight from training to testing on the larger domains. Tuning the threshold using a validation set (of larger graphs) would represent an easier problem. The model at no point, or in any way, is introduced to the data in the larger size domains for any form of training/tuning.

We decide to use the covariance matrix in this experiment, as opposed to the sample covariance matrix, as our $A_O$'s. This is for the simple reason that it would be difficult to control the snr with respect to generalization error and would be secondary to the main thrust of the experiment. When run with number of signals proportional to graph size, we see quite a similar trend, but due to time constraints, these results are not presented herein, but is apt for follow up work.

**HCP data.** HCP-YA provides up to $4$ resting state fMRI scanning sessions for each subject, each lasting 15 minutes. We use the fMRI data which has been processed by the minimal processing pipeline (Van Essen et al., 2013). For every subject, after pre-processing the time series data to be zero mean, we concatenate all available time-series together and compute the sample covariance matrix (which is what as used are $A_O$ in the brain data experiment 5.2).

Due to expected homogeneity in SCs across a healthy population, information about the SC in the test set could be leveraged from the SCs in the training set. In plainer terms, the average SC in the training set is an effective predictor of SCs in the test set. In our experiments, we take random split of the 1063 SCs into 913/50/100 training/validation/test sets, and report how much we improve upon this predictor.

Raw data available from https://www.humanconnectome.org/study/hcp-young-adult/overview.

**Co-location and social networks among high school students.** The Thiers13 dataset (Génois & Barrat, 2018) followed students in a French high school in 2013 for 5 days, recording their interactions based on physical proximity (co-location) using wearable sensors as well as investigating social networks between students via survey. The co-location study traced 327 students, while a subset of such students filled out the surveys (156). We thus only consider the 156 students who have participated in both studies. The co-location data is a sequence of time intervals, each interval

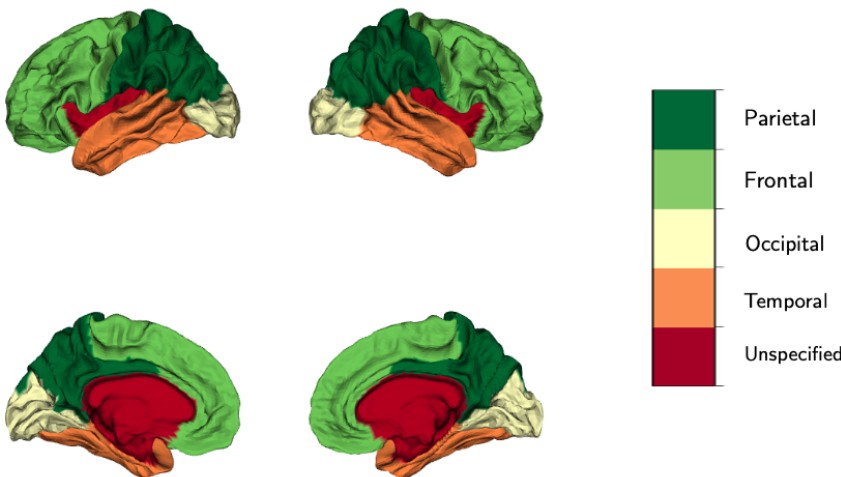

Figure 4: The four lobes in the brains cortex.

lists which students were within a physical proximity to one another in such interval. To construct the co-location network, we let vertices represent students and assign the weight on an edge between student $i$ and student $j$ to represent the (possibly 0) number of times over the 5 day period the two such students were in close proximity with one another. The social network is the Facebook graph between students: vertex $i$ and vertex $j$ have an unweighted edge between them in the social network if student $i$ and student $j$ have are friends in Facebook. Thus we now have a co-location network $\boldsymbol{A}_{O,\text{total}}$ and a social network $\boldsymbol{A}_{L,\text{total}}$. To construct a dataset of graph pairs $\mathcal{T} = \{\boldsymbol{A}_O{}^{(i)}, \boldsymbol{A}_L{}^{(i)}\}_{i=1}^{7000}$ we draw random subsets of $N = 120$ vertices (students). For each vertex subset, we construct a single graph pair by restricting the vertex set in each network to the sampled vertices, and removing all edges which attach to any node not in the subset. If either of the resulting graph pairs are not connected, the sample is not included.

The performance of the baselines on such a dataset is as follows: Threshold ($10.2 \pm 1.8e\text{-}2\%$), ND ($10.6 \pm 1.9e\text{-}2\%$), GLASSO (NA - could not converge for any of the $\alpha$ values tested), SpecTemps ($10.7 \pm 6.9e\text{-}2\%$).

## A.8 BASELINES

In the broad context of network topology identification, recent latent graph inference approaches such as DGCNN (Wang et al., 2019), DGM (Kazi et al., 2020), NRI (Kipf et al., 2018), or PGN (Veličković et al., 2020) have been shown effective in obtaining better task-driven representations of relational data for machine learning applications, or to learn interactions among coupled dynamical systems. However, because the proposed GDN layer does not operate over node features, none of these state-of-the-art methods are appropriate for tackling the novel network deconvolution problem we are dealing with here. Hence, for the numerical evaluation of GDNs we chose the most relevant baseline models that we outline in Section 5, and further describe in the sequel.

We were as fair as possible in the comparison of GDNs to baseline models. All baselines were optimized to minimize generalization error, which is what is presented in Table 1. Many baseline methods aim to predict sparse graphs on their own, yet many fail to bring edge values fully to zero. We thus provide a threshold, tuned for generalization error using a validation set, on top of each method only if it improved the performance of the method in link-prediction. The edge weights returned by the baselines can be very small/large in magnitude and perform poorly when used directly in the the regression task. We thus also provide a scaling parameter, tuned during the hyperparameter search, which provides approximately an order of magnitude improvement in MSE for GLASSO and halved the MSE for Spectral Templates and Network Deconvolution.

**Hard Thresholding (Threshold).** The hard thresholding model consists of a single parameter $\tau$, and generates graph predictions as follows

$$\hat{A}_L = \mathbb{I}\left\{|A_O| \succeq \tau \mathbf{1}\mathbf{1}^\top\right\},$$

where $\mathbb{I}\{\cdot\}$ is an indicator function, and $\succeq$ denotes entry-wise inequality. For the synthetic experiments carried out in Section 5.1 to learn the structure of signals generated via network diffusion, $A_O$ is either a covariance matrix or a correlation matrix. We tried both choices in our experiments, and reported the one that performed best in Table 1.

**Graphical Lasso (GLASSO).** GLASSO is an approach for Gaussian graphical model selection (Yuan & Lin, 2007; Friedman et al., 2008). In the context of the first application domain in Section 3, we will henceforth assume a zero-mean graph signal $x \sim \mathcal{N}(0, \Sigma_x)$. The goal is to estimate (conditional independence) graph structure encoded in the entries of the precision matrix $\Theta_x = \Sigma_x^{-1}$. To this end, given an empirical covariance matrix $A_O := \hat{\Sigma}_x$ estimated from observed signal realizations, GLASSO regularizes the maximum-likelihood estimator of $\Theta_x$ with the sparsity-promoting $\ell_1$ norm, yielding the convex problem

$$\hat{\Theta} \in \arg\max_{\Theta \succeq 0}\left\{\log\det\Theta - \text{trace}(\hat{\Sigma}_x\Theta) - \alpha\|\Theta\|_1\right\}. \tag{8}$$

We found that taking the entry-wise absolute value of the GLASSO estimator improved its performance, and so we include that in the model before passing it through a hard-thresholding operator

$$\hat{A}_L = \mathbb{I}\left\{|\hat{\Theta}| \succeq \tau\mathbf{1}\mathbf{1}^\top\right\}$$

One has to tune the hyperparameters $\alpha$ and $\tau$ for link-prediction (and a third, the scaling parameter described below, for edge-weight regression).

We used the sklearn GLASSO implementation found here: `https://scikit-learn.org/stable/modules/generated/sklearn.covariance.graphical_lasso.html`
It is important to note that we do **not** use the typical cross-validation procedure seen with GLASSO. Typically, GLASSO is used in unsupervised applications with only one graph being predicted from $s$ observations. In our application, we are predicting many graphs, *each* with $s$ observations.
Thus the typical procedure of choosing $\alpha$ using the log-likelihood [the non-regularized part of the GLASSO objective in (8)] *over splits of the observed signals, not splits of the training set*, results in worse performance (and a different $\alpha$ for each graph). This is not surprising: exposing the training procedure to labeled data allows it to optimize for generalization. We are judging the models on their ability to generalize to unseen graphs, and thus the typical procedure would provide an unfair advantage to our model. While we tried both sample covariance and sample correlation matrices as $A_O$, we found that we needed the normalization that the sample correlation provides, along with an additional scaling by the maximum magnitude eigenvalue, in order to achieve numerical stability. GLASSO can take a good amount of time to run, and so we limited the validaiton and test set sizes to an even 100/100 split. With only 2 to 3 hyperparameters to tune, we found this was sufficient (no significant differences between validation and test performance in all runs, and when testing on larger graph sizes, no difference in generalization performance).

**Network Deconvolution (ND).** The Network Deconvolution approach is "a general method for inferring direct effects from an observed correlation matrix containing both direct and indirect effects" (Feizi et al., 2013). Network Deconvolution follows three steps: linear scaling to ensure all eigenvalues $\lambda_i$ of $A_O$ fall in the interval $\lambda_i \in [-1, 1]$, eigen-decomposition of the scaled $A_O = V\text{diag}(\lambda)V^{-1}$, and deconvolution by applying $f(\lambda_i) = \frac{\lambda_i}{1+\lambda_i}$ to all eigenvalues. We then construct our prediction as $\hat{A}_L := V\text{diag}(f(\lambda))V^{-1}$. In (Feizi et al., 2013), it is recommended a Pearson correlation matrix be constructed, which we followed. We applied an extra hard thresholding on the output, tuned for best generalization error, to further increase performance. For each result shown 500 graphs were used in the hyperparameter search and 500 were used for testing.

**Spectral Templates (SpecTemp).** The SpecTemp method consists of a two-step process whereby one: (i) first leverages the model (1) to estimate the graph eigenvectors $V$ from those of $A_O$ (the eigenvectors of $A_L$ and $A_O$ coincide); and (ii) combine $V$ with a priori information about $G$ (here sparsity) and feasibility constraints on $\mathcal{A}$ to obtain the optimal eigenvalues $\lambda$ of $A_L = V\text{diag}(\lambda)V^\top$.

The second step entails solving the convex optimization problem

$$\boldsymbol{A}^*(\epsilon) := \underset{\{\boldsymbol{A},\bar{\boldsymbol{\lambda}}\}}{\operatorname{argmin}} \, \|\boldsymbol{A}\|_1, \tag{9}$$

$$\text{s. to } \|\boldsymbol{A} - \boldsymbol{V}\operatorname{diag}(\bar{\boldsymbol{\lambda}})\boldsymbol{V}^\top\|_2^2 < \epsilon, \ \boldsymbol{A}\mathbf{1} \succeq \mathbf{1}, \ \boldsymbol{S} \in \mathcal{A}.$$

We first perform a binary search on $\epsilon \in \mathbb{R}_+$ over the interval $[0,2]$ to find $\epsilon_{min}$, which is the smallest value which allows a feasible solution to (9). With $\epsilon_{\min}$ in hand, we now run an iteratively ($t$ henceforth denotes iterations) re-weighted $\ell_1$-norm minimization problem with the aim of further pushing small edge weights to 0 (thus refining the graph estimate). Defining the weight matrix $\boldsymbol{W}_t := \frac{\gamma\mathbf{11}^\top}{|\boldsymbol{A}_{t-1}^*|+\delta\mathbf{11}^\top} \in \mathbb{R}_+^{N\times N}$ where $\gamma,\delta \in \mathbb{R}_+$ are appropriately chosen positive constants and $\boldsymbol{A}_0^* := \boldsymbol{A}^*(\epsilon_{min})$, we solve a sequence $t = 1,\ldots,T$ of weighted $\ell_1$-norm minimization problems

$$\boldsymbol{A}_t^* := \underset{\{\boldsymbol{A},\bar{\boldsymbol{\lambda}}\}}{\operatorname{argmin}} \, \|\boldsymbol{W}_t \odot \boldsymbol{A}\|_1, \tag{10}$$

$$\text{s. to } \|\boldsymbol{A} - \boldsymbol{V}\operatorname{diag}(\bar{\boldsymbol{\lambda}})\boldsymbol{V}^\top\|_2^2 < \epsilon_{min}, \ \boldsymbol{A}\mathbf{1} \succeq \mathbf{1}, \ \boldsymbol{A} \in \mathcal{A}.$$

If any of these problems is infeasible, then $\boldsymbol{A}_I^*$ is returned where $I \in \{0,1,\ldots,T\}$ is the last successfully obtained solution.

Finally, a threshold $\tau$ is chosen with a validation set to map the output to binary decision over edges, namely

$$\hat{\boldsymbol{A}}_L = \mathbb{I}\left\{|\boldsymbol{A}_I^*| \succeq \tau\mathbf{11}^\top\right\},$$

We solve the aforementioned optimization problems using MOSEK solvers in CVXPY. Solving such convex optimization problems can very computationally expensive/slow, and because there are only two hyperparameters, 100 graphs were used in the validation set, and 100 in the test set.

**Least Squares plus Non-convex Optimization (LSOpt).** LSOpt consists of first estimating the polynomial coefficients in (1) via least-squares (LS) regression, and then using the found coefficients in the optimization of (3) to recover the graph $\boldsymbol{A}_L$. Due to the collinearity between higher order matrix powers, ridge regression can be used in the estimation of the polynomial coefficients. If the true order $K$ of the polynomial is not known ahead of time, one can allow $\hat{\boldsymbol{h}} \in \mathbb{R}^N$ and add an additional $\ell_1$-norm regularization for the sake of model-order selection. With $\hat{\boldsymbol{h}}$ in hand, we optimize (3), a non-convex problem due to the higher order powers of $\boldsymbol{A}$, using Adam with a learning rate of $0.01$ and a validation set of size $50$ to tune $\lambda$.

We start with the most favorable setup, using the true $\boldsymbol{h}$ in the optimization of (3), skipping the LS estimation step. Even in such a favorable setup the results were not in general competitive with GDN predictions, and so instead of further degrading performance by estimating $\hat{\boldsymbol{h}}$, we report these optimistic estimates of generalization error.

## A.9 SOURCE CODE AND CONFIGURATION

Code has been made available in a zipped directory with the submission. Refer to the README for instructions on configuring a system to run the code. We rely on PyTorch heavily and use Conda to make our system as hardware-independant as possible. GPUs were used to train the larger GDNs, which are available to use for free on Google Colab. Development was done on Mac and Linux (Ubuntu) systems, and not tested on Windows machines. The processed brain data was too large to include with the code, but is available upon request.

