# OpenReview forum: "Learning Graph Structure from Convolutional Mixtures"
_ICLR.cc/2022/Conference — ICLR 2022 Submitted_

### Official Review · Reviewer_9UsD · 2021-10-31

**Correctness:** 3
**Technical Novelty And Significance:** 4
**Empirical Novelty And Significance:** 4
**Recommendation:** 6
**Confidence:** 2

**Main Review:**

I found the paper to be well-written and easy to follow. The central ideas of the paper made sense to me and the experiments appear to validate the utility of the idea, both synthetically and in datasets with real world relevance.

Unfortunately the paper is outside my main area of expertise and therefore I am unable to make strong comments on the quality of the baseline approaches used. I will defer to the other reviewers for advice on this. But for now I'd give authors the benefit of the doubt and suggest (weak) acceptance. My understanding is that the proposed layer does not operate over node features, and as such it is not appropriate to compare it to methods like DGCNN (Wang et al., 2018), DGM (Kazi et al., 2020), NRI (Kipf, Fetaya et al., ICML'18) or PGN (Veličković et al., NeurIPS'20). Most of the above are already correctly cited in the paper.

The main concern I would have with the method as-is is that it seems to rely on an "observed" adjacency matrix ${\bf A}_O$. In many cases of interest, ${\bf A}_O$ would either not be given, or would be a result of a very crude heuristic. Would the authors' method still work if ${\bf A}_O$ were not explicitly given? If not, how could we modify the proposal to (approximately) support this? I saw no obvious discussion of this in the paper, but it is possible that I have missed something.

**Summary Of The Paper:**

The authors propose the graph deconvolutional network (GDN), which is a novel approach to graph structure recovery from noisy observed graph structures. It is based on an assumption that the observed adjacency matrix is polynomial in the true graph adjacency, and uses a proximal gradient computation to iteratively optimise for this structure. Experiments on both synthetic and brain imaging graph recovery tasks indicate the outperformance of the proposed method against several baselines.

**Summary Of The Review:**

I recommend weak acceptance because I feel like this paper presents an interesting, novel and well-grounded idea, with a decent level of evidence towards empirically ascertaining its contributions. However my confidence score is low as I lack expertise in this particular domain -- it is very much possible that I am unaware of all of the relevant baselines.

---

> ### Author Response · Authors · 2021-11-16
> **Response to Reviewer 4**
>
> Thanks for your time and effort spent in reviewing our paper, and for finding our ideas novel, significant, interesting and well grounded. We appreciate your valuable feedback and requests for clarifications on some aspects of our contribution. Next we address the issues raised, point by point.
>
> (1) On the quality of the baseline approaches used:
>
> Your assessment on the relevance of the baselines is accurate, and that same principle is what guided our choices for the experimental validation in Section  5 - Experiments. In the broad context of network topology inference, recent latent graph inference approaches (such as DGCNN, DGM, NRI, or PGN) have been shown effective in obtaining better task-driven representations of relational data for machine learning applications,  or to learn interactions among coupled dynamical systems. However, for the reason you mention none of these state-of-the-art methods  is truly appropriate for tackling the novel network deconvolution problem we are dealing with here. To give the reader a broad perspective of the advances in the field of graph learning, we nonetheless felt it was important to cite these relevant works in Subsection 1 - Related work.
>
> We acknowledge we missed the work on Pointer Graph Networks (Veličković et al., NeurIPS'20). Thanks for bringing this reference to our attention, which is now cited in the revised paper.
>
> (2) On the reliance of an observed matrix $\mathbf{A}_O$:
>
> Thanks for your comment, this is an important point. If the graph $\mathbf{A}_O$ is not explicitly given, its topology can be estimated from data and the quality of the estimate does not limit the applicability of GDNs. To fix ideas, let us consider the network neuroscience problem that motivated our development of GDNs: inferring structural brain networks from functional MRI (fMRI) signals. Therein, $\mathbf{A}_O$ is the functional connectivity (FC) network which is not directly observable, rather we estimate it as the covariance matrix of the BOLD time courses (per brain region of interest) measured with fMRI. Depending on the sample size and the number of brain regions, there will be an unavoidable estimation error in this process. In a way, we are accounting for those imperfections in the inverse problem formulation (3), by relaxing the equality constraint $\mathbf{A}_O=\mathbf{H}(\mathbf{A}_L;\mathbf{h})$. All in all, we use the term ``observed graph’’ in a broad sense (we are happy to change it if it leads to confusion), just to highlight what is given when it comes to the network deconvolution problem. In practice, it may well be the case that $\mathbf{A}_O$ is estimated from data leading to noisy observations. Brief clarifying comments along these lines have been included at the end of the revised first paragraph of Section 2 - Problem Formulation.
>
> On a related note, as we mention in Subsection 3 - Network deconvolution and denoising, another use case of GDNs could be graph denoising. Therein, $\mathbf{A}_O$ could be a corrupted graph we wish to denoise, obtained via other graph learning method upstream in the pipeline, or, some crude heuristic as you point out.
>
> In closing, if you are referring to missing entries in $\mathbf{A}_O$, then this becomes a link prediction, data imputation or matrix completion problem. In our graph learning context, one can view it as tomographic network topology inference. Of course, under some assumptions on the structure of $\mathbf{A}_O$ (say low rank if they can be modeled with random dot product graphs or stochastic block models), then one could bring to bear matrix completion algorithms as a pre-processing step to obtain an estimate of $\mathbf{A}_O$. Unsupervised link prediction modules could be tried as well. In any case, tomographic problems are very interesting and markedly harder, but well beyond the scope of this paper. We will be happy to comment on this interesting future direction in the revised paper under Section 6 - Conclusions.
>
> Thanks again for your feedback, and we look forward to addressing any follow-up concerns that you may have.

---

> > ### Comment · Reviewer_9UsD · 2021-11-22
> > **Thank you**
> >
> > Thank you for your careful response and incorporating the fixes. They are useful and will improve the clarity of the paper.
> >
> > My confidence remains relatively low, so I will prefer to retain my score for now.

---

> > > ### Author Response · Authors · 2021-11-22
> > > **Thanks!**
> > >
> > > Thanks for checking. We also believe that after incorporating the reviewer's suggestions, the revised paper has markedly improved.

---

### Official Review · Reviewer_XnFE · 2021-11-02

**Correctness:** 2
**Technical Novelty And Significance:** 2
**Empirical Novelty And Significance:** 2
**Recommendation:** 5
**Confidence:** 2

**Main Review:**

While the overall idea and the algorithm part of the paper seems to be ok. There is a  fundamental question that need to be addressed. In (1), there is no constraints on alphas and As, in this case, is the solution A in (1) unique? If it is unique, then we can conclude that the problem the author are trying to attack is identifiable.

Or if there are multiple A that satisfies (1) with a given A_O, is the skeleton of A unique? i.e. regardless of the weight, just consider the unweighted version of A. If this uniqueness can be established, then the identifiable can also be established.



**Summary Of The Paper:**

The authors tries to recover the underlying graph structures from observed symmetric adjacency matrix. The key assumption of the paper is that the observed adjacency matrix can be represented as a polynomial of the adjacency matrix of the true underlying graph, which is reasonable. The experiments show that the proposed model can recover graph structures on provided dataset.

**Summary Of The Review:**

One fundamental question about the paper is not clearly stated.

---

> ### Author Response · Authors · 2021-11-18
> **Response to Reviewer 3**
>
> Thanks for your time and effort spent in reviewing our paper, and for finding our ideas and algorithm satisfactory as well as the modeling assumptions reasonable. We appreciate you bringing up the fundamental question about model identifiability, which we address next.
>
> On the fundamental question of model identifiability:
>
> This is a very interesting point you raise here. Without any constraints on $\mathbf{h}$ and $\mathbf{A}_L$, the problem of recovering $\mathbf{A}_L$ from $\mathbf{A}_O=\mathbf{H}(\mathbf{A}_L;\mathbf{h})$ as in (1) is clearly non-identifiable. Indeed, if the desired solution is $\mathbf{A}_L$ (with associated polynomial coefficients $\mathbf{h}$), there is always at least another solution $\mathbf{A}_O$ corresponding to the identity polynomial mapping. This is why adding structural constraints like sparsity on $\mathbf{A}_L$ (as we do in Section 2), will aid model uniqueness/identifiability, especially when devoid of training examples. Moreover, being an adjacency matrix of an undirected graph $G$, $\mathbf{A}_L$ has hollow diagonal, non-negative edge weights and is symmetric, thus it is subject to the constraints $\mathbf{A}_L\in \mathcal{A}$, where $\mathcal{A}$ is the convex admissibility set defined after (3).
>
> It is worth mentioning that (1) implies the eigenvectors of $\mathbf{A}_L$ and $\mathbf{A}_O$ coincide. So the eigenvectors of the sought latent graph are given once we observe $\mathbf{A}_O$, what is left to determine are the eigenvalues. We have in practice observed that for several families of *sparse, weighted graphs* the eigenvector information along with the constraint $\mathbf{A}_L\in \mathcal{A}$ are sufficient to uniquely specify the graph (meaning a sufficient rank-related uniqueness condition we derive is satisfied). Interestingly, this implies that many random weighted graphs can be uniquely determined from their eigenvectors. This strong uniqueness result does not render our problem vacuous, since seldomly in practice one gets to observe $\mathbf{A}_O$ (and hence its eigenvectors) error free; see also our response to comment (2) by Reviewer 4.
>
> If one were to formally study identifiability of this problem (say under some structural assumptions on $\mathbf{A}_L$ and/or the polynomial mapping), then one has to recognize the problem suffers from an inherent scaling ambiguity. Indeed, if given $\mathbf{A}_O=\mathbf{H}(\mathbf{A}_L;\mathbf{h})$ (which means the pair $\mathbf{A}_L$ and $\mathbf{h}=[h_0,h_1,\ldots,h_K]^\top$ is a solution), then for any positive scalar $\alpha$ one has that $\alpha\mathbf{A}_L$ and $[h_0,h_1/\alpha,\ldots,h_K/(\alpha^K)]^\top$ is also a solution. Accordingly, uniqueness claims can only be meaningful modulo this unavoidable scaling ambiguity.
>
> But *this ambiguity is lifted once we tackle the problem in a supervised learning fashion* -- our approach in this paper. The training samples in $\mathcal{T}$ fix the scaling, and accordingly the GDN can learn the mechanism or mapping of interest $\mathbf{A}_O \mapsto \mathbf{A}_L$. Hence, an attractive feature of the GDN approach is that by using data, some of the inherent ambiguities in (1) are naturally overcome. In particular, the SpecTemp approach in (Segarra et al., Network topology inference from spectral templates. *IEEE Trans. Signal Inf. Process. Netw.*, 2017) relies on convex optimization and suffers from this scaling ambiguity, so it requires an extra (rather arbitrary) constraint to fix the scale. The network deconvolution approach in (Feizi et al., Network deconvolution as a general method to distinguish direct dependencies in networks. *Nat. Biotechnol*, 2013) relies on a fixed, known polynomial mapping, and while it does not suffer from these ambiguities it is limited in the graph convolutions it can model.
>
> All in all, the inverse problem associated to (1) is just our starting point to motivate a trainable parametrized architecture $\hat{\mathbf{A}}_L=\Phi(\mathbf{A}_O;\mathbf{\Theta})$, that introduces valuable inductive biases to generate graph predictions. The problem we end up solving is different (see the formal statement in Section 2) because we rely on supervision using graph examples, thus rendering many of these fascinating uniqueness questions less relevant.  We plan to include these discussion points on uniqueness/identifiability in an Appendix of the revised manuscript. If you believe certain arguments are worthy of the main paper, we will be happy to accommodate that as well.
>
> Thanks again for your feedback, and we look forward to addressing any further concerns that may arise.

---

> > ### Comment · Reviewer_XnFE · 2021-11-26
> > **Identifiability**
> >
> > The identifiability of the problem is very important. Even if in a learning-based manner. If the identifiability can not be established, how can you confirm the feed in data is correct? E.g. for the same input, if there exists two different solution, you can make two conflict training sample to the neural network and the neural network can break down. The authors should at least try to establish identifiability on some equivalent class, e.g., treat A_L with different scale as the same.

---

> > > ### Author Response · Authors · 2021-11-26
> > > **Follow-up on identifiability**
> > >
> > > We respectfully point out that the question of identifiability is inmaterial to our paper. The identification of topology formulated in our paper is under-specified. There are several possible graphs that can generate the observed signals.
> > >
> > > Regarding the training data being correct, as per the problem statement samples are given (brain functional and structural connectivities, co-locations networks and Facebook friendships) and we want to learn a mapping for these data. It is straightforward to check we are not training with conflicting samples that have a common $\mathbf{A}_O$ and respectively different $A_L$s, so we respectfully believe this envisioned scenario is not limiting in practice.

---

### Official Review · Reviewer_CCxm · 2021-11-02

**Correctness:** 4
**Technical Novelty And Significance:** 3
**Empirical Novelty And Significance:** 2
**Recommendation:** 5
**Confidence:** 5

**Main Review:**

**Strengths:**

This work formulates the graph structure learning problem as an inverse problem. Such a formulation is interesting and it differs from many of the latest formulations based on deep learning.

**Weaknesses:**

However, there are multiple concerns on the modeling, the setting, and the evaluation.

**Model and misnomer of naming.** The authors posit a polynomial model $x = H(A_L , h) w$, where $H$ is a polynomial of $A_L$, and name the overall approach a "graph deconvolution network" (GDN). Such a naming allures a reader to draw direct analogy with graph convolutional networks (GCN). However, there is a critical difference between GCN and GDN, which brings more confusion than appreciation. If we consider $x = H(A_L , h) w$ a graph convolution, then a key of GCN is the learnable transformation on each component of the input signal $w$, lacking in GDN. A graph convolution, either interpreted as a filter or as a neighborhood aggregator, has a limited capability in generating output signals, without such learnable transformations. This is not to say that the authors' polynomial model is wrong; the model is just limited and calling the approach GDN causes big confusion.

**Setting.** It is a bit unclear if the assumption of known $(A_O, A_L)$ pairs is reasonable. This assumption, of course, allows supervised learning and it also appears to be supported by the neuroimaging application the authors demonstrate. On the other hand, the point of the application may need more articulation to be convincing. The sole purpose appears to learn brain structural connectivity, but what is the point of learning it? The authors have not given clues on the use of the structure or insights obtained therein. Moreover, is the structural connectivity manually annotated or algorithmically annotated? If the latter, why machine learning?

Going beyond, the paper may benefit from suggesting more applications of inferring $A_L$ from $A_O$, where supervised data are available. The reviewer is a bit skeptical that the supervised setting has very limited use.

**Evaluation.** There is a straightforward baseline and the authors are encouraged to compare with it, too. One may use the training data to estimate the coefficients $h$ of the polynomial $H(A,h)$. Once these coefficients are obtained, one can estimate $A_L$ for any given $A_O$. The first step is simply a least squares problem and the second step can be solved by using proximal gradient. Both are easy to solve. Furthermore, one may let $h$ have $N$ entries and use a regularization to estimate them, leading to again a proximal gradient solution.

**Minor comments:**

- Section 4.2. The loss is an important piece of information. Currently the main text is a bit cryptic and the details are deferred to the appendix. The authors should at least describe the losses (if not giving formulas) in the main text.

- If the authors draw connections between an RNN and the GDN, then the initial $A[0]$ serves a similar role to the initial input to RNN. An additional approach is to treat it a learnable parameter.

---

**After rebuttal**

This paper has technical merits and the revisions done during the rebuttal period solidify the work. In many aspects, the authors successfully convince me of the value of the setting and the applications, although the naming of the model as "graph deconvolution network" will likely confuse researchers and practitioners in the field of graph neural networks. I encourage the authors to complete the remaining experimental evaluations. The added Facebook experiment provides a good application and I feel it may be even more intuitive than the brain structure application the authors originally focus on, depending on the background of the readers.


**Summary Of The Paper:**

This paper proposes an approach to estimate a latent graph $A_L$ given an observed graph $A_O$ (e.g., the covariance matrix of signals generated through a graph diffusion process). The authors posit a polynomial model and unroll proximal gradient iterations to estimate a variant of the model. The authors conduct experiments on synthetic datasets with supervised $(A_O, A_L)$ pairs as well as a neuroimaging dataset, where $A_O$ denotes the functional connectivity graph and $A_L$ denotes the structural connectivity graph. The proposed approach performs better than several baseline approaches.

**Summary Of The Review:**

This work formulates the graph structure learning problem as an inverse problem. Such a formulation is interesting. However, there are multiple concerns on the modeling, the setting, and the evaluation, which leave ample room for improvement before the paper can be published.

---

> ### Author Response · Authors · 2021-11-17
> **Response to Reviewer 2 - Part 1**
>
> Thank you for your thoughtful feedback as well as for finding the inverse problem formulation interesting and different from most recent graph structure identification approaches based on deep learning. We address the various concerns raised in the following point-by-point responses.
>
> (1) On the model and the name of the proposed architecture:
>
> Our rationale for adopting the name Graph Deconvolution Network (GDN) is that we propose a neural network-based learning approach to tackle a *network deconvolution problem.*
>
> We acknowledge your concern on potential confusion caused if the reader draws an analogy between GDNs and GCNs from the existing literature. To address this, in the revised paper we clarify that the assumed data model could further be expanded to improve expressivity for a wider range of output signals and better relate the convolutional data model to GCNs. For instance, the parameters $\mathbf{h}$ could be learned using existing data-driven, learning approaches. However, these aspects of the data model are neither central to the problem considered nor do they pose any unique technical challenges and therefore, can be readily accommodated in the assumed data model for the GDN framework. We also remark that the convolutional data model in its given form has been adopted in existing literature in a wide range of studies, as summarized in Section 3 - Motivating Application Domains.
>
> To further elaborate on the relevance and breadth of applicability of the convolutive (or network diffusion) signal model $\mathbf{x}=\mathbf{H}(\mathbf{A}_L;\mathbf{h})\mathbf{w}$, we would like to elucidate connections with related work for graph structure identification. Note that while we used the diffusion-based generative model for our derivations in Subsection 3 - Latent graph structure identification from diffused signals, we do not need it as a mechanistic process. Indeed, the only thing we ask in (1) is for the data covariance $\mathbf{A}_O=\mathbf{\Sigma}_x$ to be some analytic function of the latent graph $\mathbf{A}_L$.  Many workhorse statistical methods for topology inference (implicitly) make specific choices for such an analytic mappings, e.g. (i) correlation networks rely on the identity mapping $\mathbf{\Sigma}_x=\mathbf{A}_L$; (ii) Gaussian graphical model selection methods, such as graphical lasso, adopt  $\mathbf{\Sigma}_x=\mathbf{A}_L^{-1}$; and (iii) undirected structural equation models $\mathbf{x}=\mathbf{A}_L\mathbf{x}+\mathbf{w}$ which implies $\mathbf{\Sigma}_x=(\mathbf{I}-\mathbf{A}_L)^{-2}$. These models are subsumed by the general framework we put forth here. We will be happy to include this discussion in a short appendix, if you believe (as we do) it contributes towards better conveying the relevance and generality of the proposed model and GDN to tackle the graph learning task.
>
> (2) On the relevance of the problem setting:
>
> Thank you for this valuable feedback. Functional connectivity (FC), constructed using fMRI, and structural connectivity (SC), constructed using dMRI, require multiple scans and computationally expensive processing pipelines (e.g. QSIPrep for SC, fMRIPrep for FC) to obtain the respective network graphs. The dMRI processing pipelines producing SC are particularly fraught due to quality issues in the data, see e.g. the recent review (Yeh et al., Mapping structural connectivity using diffusion MRI: Challenges and opportunities. *J Magn Reson Imaging*, 2021), and can take a long time, for instance QSIPrep may take up to 18 hours to extract SC for a single subject in the HCP dataset. The ability to collect only FC and get informative estimates of SC open the door to large scale studies, previously constrained by the logistical, cost, and computational resources needed to collect both modalities. Additionally, the structure-function coupling is a feature of interest in studying neurological diseases since it is known to vary with respect to healthy subjects in pathological contexts. Our experiments in Section 5.2 - HCP Brain Data show the value of GDNs in studying the structure-function relationship in the brain. Therefore, our results could potentially serve as baselines that characterize healthy subjects, and expanding our work to study the deviations in findings on similar data in a pathological context would be of immediate interest. To the best of our knowledge, there is no known deterministic, invertible analytical model that allows recovery of SC from FC and therefore, GDNs offer a viable approach to learn this mapping. General remarks along these lines have been included in the revised Subsection 3 - Inferring structural brain networks from functional MRI (fMRI) signals, to make a convincing case on the relevance of this problem that motivated our development of GDNs.

---

> > ### Comment · Reviewer_CCxm · 2021-11-18
> > **Follow up the response**
> >
> > Dear authors, your discussions clear several of my doubts and are much appreciated. I do not seem to see an updated paper; I intend to adjust the score when I see these discussions entering the paper.
> >
> > The simple baseline I suggested is important because you motivate the method from a polynomial model, for which a straightforward solution needs be compared. The second step in the baseline is a nonconvex problem and proximal gradient indeed may not work well. I imagine solving it like general nonconvex programming for neural networks: apply Adam with L1 regularization in PyTorch (thus no need to supply gradient). Of course, other optimization libraries may also do the job.

---

> > > ### Author Response · Authors · 2021-11-20
> > > **Updated paper uploaded**
> > >
> > > Thanks for following up and for providing valuable additional feedback. We are glad to hear that our responses and discussions addressed your initial concerns. Your suggestions (along with those of the other reviewers) have led to a much improved revised manuscript. The updated paper is now available in the system, and the major changes implemented are color-coded blue for ease of checking.  We believe this work proposes a new and interesting problem formulation, while the algorithmic solution offers a significant contribution to the field of graph structure identification with tangible impact to applications.
> > >
> > > Some noteworthy major points in the revision include:
> > > - A discussion on model identifiability is included in Appendix A.1, as requested by Reviewer 3.
> > > - Following your suggestions, we elaborate on the importance of the network neuroscience problem (both in Sections 3 and 5.2) and suggest more application domains where predicting $\mathbf{A}_L$ from $\mathbf{A}_O$ using supervised data is well motivated (Section 3).
> > > - Along the lines of the previous comment and following the suggestion of Reviewer 1, in Section 5.2 we now include a brand new real-data experiment on using GDNs to predict Facebook ties using human co-location data (additional details are in Appendix A.7).
> > > - In Section 5.1 we included ablation studies to assess the effect of: (i) the prior $\mathbf{A}[0]$ (e.g., when it is learned given the links between RNNs and GDNs); and (ii) the truncation of the gradient $\nabla g(\mathbf{A})$ when we derive GDNs. Both of these points were valuable suggestions by Reviewer 1.
> > > - To further elaborate on the relevance and breadth of applicability of the graph convolutional (or network diffusion) signal model $\mathbf{x}=\mathbf{H}(\mathbf{A}_L;\mathbf{h})\mathbf{w}$, we have included discussions in Appendix A.2 to better place it context of other workhorse graph learning approaches.
> > >
> > > We are still working on the paper, in particular we are running the two-step baseline you suggested which we describe as: "...least-squares fitting of $\mathbf{h}$ followed by non-convex optimization to solve (3) (LSOpt)..." This experiment is time consuming because we have to run it over all the graph families in Table 1 (RG, ER, BA, SC), over different graph filter realizations, and in each case perform a cross-validation procedure to tune the regularization parameter in (3). Our initial observations suggest that performance is not great, barely better than Hard Thresholding (Threshold). We look forward to having those results included in Table 1 by the deadline.
> > >
> > > Thanks again for your feedback, and we look forward to addressing any further concerns that may arise.

---

> > > > ### Comment · Reviewer_CCxm · 2021-11-22
> > > > **Score raised**
> > > >
> > > > I left some feedback in the "Main Review" section.

---

> > > > > ### Author Response · Authors · 2021-11-22
> > > > > **Thanks!**
> > > > >
> > > > > We appreciate your feedback. Glad to hear you liked the Facebook experiment and found it intuitive. We have completed the experimental evaluation, and the results for the LSOpt baseline you suggested can be found in Table 1. We report that even in the most favorable setting whereby the ground-truth polynomial coefficients are fed to the second step [non-linear optimization to tackle the inverse problem (3)], GDN outperforms LSOpt in terms of error and MSE. Details are in Appendix A.8.
> > > > >
> > > > > Looking forward to your final assessment and hopefully a score above the acceptance threshold. If there are any outstanding comments that you believe can further improve the paper, we will be happy to implement those in the final hours of the discussion window.

---

> ### Author Response · Authors · 2021-11-17
> **Response to Reviewer 2 - Part 2**
>
> Your suggestion on discussing more applications to better justify a wider applicability of GDNs is much appreciated. To this end, in the revised paper we elaborate on other potential applications where supervised data is available from bioinformatics [such as inferring protein contact structure from mutual information graphs of the covariation of amino acid residues; see (Feizi et al., Network deconvolution as a general method to distinguish direct dependencies in networks. *Nat. Biotechnol*, 2013)], social and information networks [say graph sparsification to unveil the most relevant collaborations in a social network encoding co-authorship information; see e.g., (Spielman & Srivastava, Graph sparsification by effective resistances. *SIAM Journal on Computing*, 2011)], and epidemiology (such as contact tracing by deconvolving the graphs that model observed disease spread in a population). We are currently working on an additional real data experiment and will provide updates on the results as soon as we get them.
>
> (3) On the evaluation:
>
> Thanks for your suggestion of this baseline. We have also thought about this natural two-step approach, but opted not to report it in the original submission. The reason being that the second step after estimating the filter coefficients entails a *non-convex* problem (notice the polynomial dependence on $\mathbf{A}_L$ of the resulting inverse problem). So even using a proximal gradient method, the resulting problem is arguably not that easy to solve. Instead, among the relevant two-step baselines suitable for the network deconvolution problem, in the original paper we decided to implement and compare against the spectral templates (SpecTemp) algorithm of (Segarra et al., Network topology inference from spectral templates. *IEEE Trans. Signal Inf. Process. Netw.*, 2017).  The approach therein is to first estimate the eigenvectors of $\mathbf{A}_L$ [those are shared with $\mathbf{A}_O$ under the model (1)], and then estimate the eigenvalues of $\mathbf{A}_L$ by solving a sparsity-regularized *convex* problem. The results in Table 1 show that GDN outperforms SpecTemp uniformly across the random graph ensembles we tested.
>
> All this being said, we will be happy to implement and compare against the suggested baseline if you still believe the results obtained will improve the performance evaluation protocol of our paper. **We wanted to clarify that point and get your input before proceeding.**  We value your input to improve the quality of our work and better highlight its merits.
>
> (4) On the minor issues raised:
>
> Following your suggestion, in the revised Sections 2 and 4.2 we now describe the loss functions chosen for the edge-weight regression and link prediction tasks, while detailed formulas are deferred to the Appendix A.2. Specifically, in Section 2 the paper now reads: “The loss $\ell$ is chosen to accommodate the task at hand -- hinge loss for link prediction or mean-squared/absolute error for the more challenging edge-weight regression problem; see Appendix A.2.”
>
> Moreover, we appreciate your valuable idea to learn the prior $\mathbf{A}[0]$. Considering also the feedback provided by Reviewer 1, in the revised paper we have conducted an ablation study to assess the effect of the prior $\mathbf{A}[0]$ on the recovery performance. In the context of recovering random graphs adhering to a stochastic block model, we tested various fixed priors and also experimented with learning $\mathbf{A}[0]$ as you suggested. As expected, a learnt prior yielded the best performance; see also our response to Reviewer 1 for additional details on the ablation study. Moreover, in the revised Subsection 4.3 - Incorporating prior information via algorithm initialization, we have added the following brief remark "Recalling the connections drawn between GDNs and RNNs, then the prior $\mathbf{A}[0]$ plays a similar role to the initial RNN input and thus it could be learned.’’
>
> Thanks again for your feedback, and we look forward to addressing any further concerns that may arise.

---

### Official Review · Reviewer_ADij · 2021-11-04

**Correctness:** 3
**Technical Novelty And Significance:** 3
**Empirical Novelty And Significance:** 3
**Recommendation:** 5
**Confidence:** 4

**Main Review:**

Strengths:
The problem on graph structure inference is not novel, yet the authors provided an interesting aspect by formulating it with unrolling algorithm, which is then solved using truncated proximal projection. Some important tricks are also introduced to enhance the performance of GDN.
Weaknesses:
(1) Some claims and descriptions are confusing, as listed below:
1. In abstract, ... assuming the knowledge of said graphs may be untenable in practice .... This claim is too bold. There are quite a lot of well-explored tasks have reasonable good graph structure, like molecular graph[1].
2. In Sec 4.1, why PG iterations (Eq 4) is sufficient to solve Eq 3? Though it can be implied from the equation, where $k$ denotes the iterations; the authors may explicitly add intuitions here, especially for audience not familiar with the optimization methods.
3. In Sec 4.2, why can authors drop all higher-order terms for simplicity? This may benefit for computation, yet how about the approximation error? The community has acknowledged the Taylor expansion with higher-order truncation, but I'm not sure how this would affect the Eq 5. Maybe some previous work (if exists) or some textual explanations can be added here.

(2) The role of prior knowledge in GDN. The initial selection of A[0] seems to have an important effect on the performance. In Sec 4.2, it says ... we can select to incorporate prior information ..., and Sec 4.3 lists some special cases. However, there does not seem to be an universal optimal solution. I would expect the authors to have an ablation to test what's the effect of different priors on A[0], and cases with K=0.
(3) About empirical results. I'm not familiar with the literature, so cannot judge if there are key baselines missing or if the empirical performance gain is substantial. But according to this paper:
1. In Sec 1, related work part, the authors also introduce more recent works, like [2,3] and more, but they are not included in Sec 5. Even though they have some drawbacks (lack of robustness, scalability issue, etc) as stated in the paper, including them can better support the effectiveness of GDN. Any idea why they are excluded?
2. In Sec 5, there's only one real dataset. Is this sufficient to prove the effectiveness of GDN in the general setting?

-----
[1] Wu, Zhenqin, et al. "MoleculeNet: a benchmark for molecular machine learning." Chemical science 9.2 (2018): 513-530.
[2] Xiaowen Dong, Dorina Thanou, Michael Rabbat, and Pascal Frossard. Learning graphs from data: A signal representation perspective. IEEE Signal Process. Mag., 36(3):44–63, 2019.
[3] Bastien Pasdeloup, Vincent Gripon, Gregoire Mercier, Dominique Pastor, and Michael G. Rabbat. Characterization and inference of graph diffusion processes from observations of stationary signals. IEEE Trans. Signal Inf. Process. Netw., 4(3):481–496, 2018.


**Summary Of The Paper:**

This paper proposes a novel solution to infer the graph structure. It starts with the unrolling algorithm for graph structure inference problem, and then adopt the proximal gradient iterative solutions. The expressiveness is augmented by parameterized with deep neural network, called Graph Deconvolution Network (GDN).


**Summary Of The Review:**

The proposed algorithm in this paper sounds technically interesting. However, I have some concerns on some of the descriptions, model designs, and empirical results. Thus, for the current version, I would rate it as borderline paper, and I will consider raising the score if the main concerns are properly solved during the discussion period.

---

> ### Author Response · Authors · 2021-11-17
> **Response to Reviewer 1 - Part 1**
>
> Thanks for your time and effort spent in reviewing our paper, and for finding the solution novel and the proposed algorithm technically interesting. We appreciate your suggestions and requests for clarifications on various aspects of our work, which we address below. We believe these constructive comments have led to an improved revised paper.
>
> (1) On the confusing claims and descriptions:
>
> 1- This point is well taken. Following your suggestion, we have recalibrated the claim in the abstract so that it now reads: “However, when said graphs are (partially) unobserved, noisy, or dynamic, the problem of inferring graph structure from data becomes relevant.”
>
> 2-  Proximal gradient methods are well suited to tackle problems with composite objective functions, when one of the component functions is differentiable and the other one admits a proximal operator that can be evaluated efficiently (here, the $\ell_1$ norm leading to a soft-thresholding operator or ReLU). Even in the non-convex setting, we believe the iterations in (4) capture enough of the structure in the problem and serve as an algorithmic blueprint that we unroll to a neural network architecture. This way, said structure translates to favorable inductive bias to aid learning. Following your suggestion, in the revised paper we added further intuitions to help those readers not well-versed in optimization. Specifically, after (4) we indicate that the PG iteration boils down to a gradient descent step followed by the proximal operator of the $\ell_1$ norm; that is the ReLU. We also direct the interested reader to the review paper (Parikh & Boyd, Proximal Algorithms. *Foundations and Trends in Optimization*,  2014) for additional background.
>
> 3- Good question. The practical reason the higher-order gradient terms are dropped is for training stability. We experimented with higher-order terms $(K>1)$, and did not achieve stable training. As we mention in the paper, an affine approximation leads to more benign optimization landscapes (else parameters in one layer would enter polynomially in subsequent layers, challenging optimization). Other methods, like GCNs (Kipf & Welling, ICLR’17), use a similar approach for their architectural design. The rationale is that one can make up for losses in per-layer expressivity by stacking layers, as we do. The ability of GDNs to perform well on synthetics, and the HCP dataset, shows we indeed retain expressivity with stacked layers.
>
> Following your suggestion, as part of the new ablation study we have also tested the architecture resulting from choosing $K=0$, hence further truncating the gradient. Lacking a linear term that effectively facilitates information aggregation in the graph, our results show that in this case the model is not expressive enough and performance markedly degrades (link-prediction percentage error of $25.72\pm1.3e\text{-}2$ and MSE of $1.7e-1±4.7e\text{-}4$  in recovering random geometric graphs (RG); see Table 1 in the original submission for the performance of GDNs with $K=1$).
>
> (2) On the role of the prior $\mathbf{A}[0]$ in the GDN model:
>
> This is a very good point. Indeed, the choice of $\mathbf{A}[0]$ has an effect on performance, and as mentioned by Reviewer 2 it even makes sense to learn the prior (given the relationship between unrolled architectures -- such as GDNs -- and RNNs where one often learns the initial state). Following your suggestion, we conducted an ablation study to assess the effect of the prior, and the relevant piece of the new Subsection 5.1 - Ablation studies now reads:
>
> "The choice of prior can influence model performance, as well as reduce training time and number of parameters needed. When run on stochastic block model (SBM) graphs with $N=21$ nodes and $3$ equally-sized communities (within block connection probability of $0.6$, and $0.1$ across blocks),  for the link-prediction task GDNs attain a percentage error of $16.8 \pm 2.7e\text{-}2$,  $16.0 \pm 2.1e\text{-}2$, $14.5 \pm 1.0e\text{-}2$, $14.3 \pm 8.8e\text{-}2$ using an all zeros, all ones, block diagonal, and learned prior, respectively. The performance improves when GDNs are given an informative prior (here a block diagonal matrix corresponding to the communities in the SBM graphs), and improves further when GDNs are allowed to learn the prior $\mathbf{A}[0]$.’’

---

> > ### Author Response · Authors · 2021-11-28
> > **Follow up**
> >
> > Thanks again for your evaluation of our paper. We hope you have had a chance to examine our response and the revisions made to the paper, which also include the results on another real dataset in Section 5.2 as promised in our earlier response. This is a follow up to inquire if your concerns have been appropriately addressed and if so, we'd be grateful if you re-evaluate your rating of the paper.

---

> ### Author Response · Authors · 2021-11-17
> **Response to Reviewer 1 - Part 2**
>
> (3) On the empirical evaluation and results:
>
> 1- In the broad context of network topology inference, recent latent graph inference approaches (such as DGCNN, DGM, NRI, or PGN) have been shown effective in obtaining better task-driven representations of relational data for machine learning applications, or to learn interactions among coupled dynamical systems. However, because the proposed GDN layer does not operate over node features, none of these state-of-the-art methods is appropriate for tackling the novel network deconvolution problem we are dealing with here. To give the reader a broad perspective of the advances in the field of graph learning, we nonetheless felt it was important to cite these relevant works in Subsection 1 - Related work. See also the related comment by Reviewer 4.
>
> Specifically with regards to (Dong et al., Learning graphs from data: A signal representation perspective. IEEE Signal Process. Mag., 2019), this is a survey paper on recent advances in graph learning. Most of the relevant methods therein have been already implemented as baselines, namely graphical Lasso, network deconvolution, and spectral templates (SpecTemp). The reference (Pasdeloup et al., Characterization and inference of graph diffusion processes from observations of stationary signals. IEEE Trans. Signal Inf. Process. Netw., 2018) is a particular case of the SpecTemp approach, which we already included as part of our performance evaluation protocol.
>
> 2- Following your suggestion on expanding our evaluation on real datasets, we are currently working on additional experiments and will provide updates on the results as soon as we get them.
>
> Thanks again for your feedback, and we look forward to addressing any further concerns that may arise.

---

### Decision · Program_Chairs · 2022-01-20

**Decision:**

Reject

**Comment:**

The paper addresses the problem of recovering a graph structure from empirical observations. The proposed approach consists of formulating the problem as an inverse problem, and then unrolling a proximal gradient descent algorithm to generate a solution.

Whereas the paper has definitely some merit, it received borderline reviews, with three borderline rejects and one borderline accept. The reviewers have appreciated the clarifications and discussions provided by the rebuttal, and one reviewer went up from reject to borderline reject. More precisely, this reviewer agrees that the paper has become stronger, but he/she believes that the paper requires additional experimental work (see section "After rebuttal" from his/her review). Another active reviewer during the rebuttal/discussion stage was not convinced by the rebuttal, after raising issues about identifiability. The area chair agrees that solving the identifiability issue is not a key requirement for this paper; however, this raises legitimate questions about the guarantees/properties of the returned solutions.

Overall, this is a borderline paper, which introduces an interesting idea, but which requires additional experimental work and discussions about the properties of the solutions. Unfortunately, the area chair agrees with the majority of the reviewers and follows their recommendation. The two previous points should be addressed if the paper is resubmitted elsewhere.

---

> ### Public Comment · ~Max_Wasserman1 · 2022-02-17
> **Authors’ Response to Paper Decision**
>
> We would like to thank the Area Chair for handling our submission and for confirming that the question of identifiability is inmaterial to our paper. We are disappointed to hear that the main reason for rejection is that the paper requires additional experimental work, citing the section “After rebuttal” in the comments of one of the reviewers. This comment explicitly says, “I encourage the authors to complete the remaining experimental evaluations” and is in response to us previously stating “We are still working on the paper, in particular we are running the two-step baseline you suggested….We look forward to having those results included in Table 1 by the deadline.” We want to state that we did complete all requested experimental work by the submission deadline, and the LSOpt baseline results are tabulated under Table 1 (two lines of results color coded blue). It is unfortunate this was missed by the Area Chair when making the final recommendation.
>
> In closing, we would like to acknowledge the reviewers for their valuable feedback and constructive suggestions that led to an improved revised paper.